# A role for CIM6P/IGF2 receptor in memory consolidation and enhancement

**Xiao-Wen Yu, Kiran Pandey, Aaron C Katzman, Cristina M Alberini\***

Center for Neural Science, New York University, New York, United States

**Abstract** Cation-independent mannose-6-phosphate receptor, also called insulin-like growth factor two receptor (CIM6P/IGF2R), plays important roles in growth and development, but is also extensively expressed in the mature nervous system, particularly in the hippocampus, where its functions are largely unknown. One of its major ligands, IGF2, is critical for long-term memory formation and strengthening. Using CIM6P/IGF2R inhibition in rats and neuron-specific knockdown in mice, here we show that hippocampal CIM6P/IGF2R is necessary for hippocampus-dependent memory consolidation, but dispensable for learning, memory retrieval, and reconsolidation. CIM6P/IGF2R controls the training-induced upregulation of *de novo* protein synthesis, including increase of Arc, Egr1, and c-Fos proteins, without affecting their mRNA induction. Hippocampal or systemic administration of mannose-6-phosphate, like IGF2, significantly enhances memory retention and persistence in a CIM6P/IGF2R-dependent manner. Thus, hippocampal CIM6P/IGF2R plays a critical role in memory consolidation by controlling the rate of training-regulated protein metabolism and is also a target mechanism for memory enhancement.

**\*For correspondence:**
ca60@nyu.edu

**Competing interests:** The authors declare that no competing interests exist.

## Introduction

CIM6P/IGF2R, a type one integral membrane glycoprotein with a short cytoplasmic tail and 15 extracytoplasmic repeats, binds to a variety of ligands, among which the better characterized belong to two groups that bind to distinct sites: glycoproteins conjugated with mannose-6-phosphate (M6P), including lysosomal enzymes, and IGF2 (*Ghosh et al., 2003*; *Morgan et al., 1987*; *Wang et al., 2017*). CIM6P/IGF2R is implicated in numerous cellular processes including growth, trafficking of lysosomal enzymes, lysosome-mediated clearance of the polypeptide IGF2, and proteolytic activation of enzymes and growth factor precursors (*El-Shewy and Luttrell, 2009*; *Ghosh et al., 2003*; *Hawkes and Kar, 2004*; *Wang et al., 2017*). CIM6P/IGF2R traffics newly synthesized lysosomal enzymes from the trans-Golgi network to lysosomes via early and late endosomes, where the enzymes are released in the acidic environment; late endosomes then fuse with lysosomes, while the CIM6P/IGF2R recycles back to the Golgi or to the cell surface via endosomal transport. CIM6P/IGF2R expressed on the cell membrane is best known for sequestering and internalizing IGF2, leading to its degradation in lysosomes (*Oka et al., 1985*); in this case as well, the receptor recycles back to its intracellular or plasma membrane compartments. The same trafficking pathway is also used by a subset of lysosomal enzymes that are secreted into the extracellular space: the enzymes bind to cell surface CIM6P/IGF2R, and are then targeted to lysosomes; see *Ghosh et al. (2003)*, for a thorough review of the binding partners and sorting signals for CIM6P/IGF2R trafficking. CIM6P/IGF2R-null mice exhibit perinatal lethality, significant overgrowth abnormalities in organ development, particularly in cardiac tissue, and mis-sorting of M6P-tagged lysosomal enzymes. Collectively, these phenotypes indicate that CIM6P/IGF2R is crucial for normal growth and development, as well as for the correct trafficking of lysosomal enzymes (*Lau et al., 1994*; *Ludwig et al., 1996*); Z.-Q. *Wang et al. (1994)*; *Wylie et al., 2003*).

Outside of the developmental context, CIM6P/IGF2R plays important roles in growth inhibition, including tumor suppressor functions. In many types of human cancers, including tumors of the lung,

liver, and breast, CIM6P/IGF2R frequently undergoes loss of heterozygosity, often accompanied by loss-of-function mutation in the remaining allele (*De Souza et al., 1995*; *Hankins et al., 1996*; *Kong et al., 2000*; *Ouyang et al., 1997*). Loss of CIM6P/IGF2R function is thought to allow abnormally high levels of circulating IGF2 to exert mitogenic effects, presumably acting via its low-affinity receptor IGF1R (*Ludwig et al., 1996*), resulting in tumor growth.

Notably, CIM6P/IGF2R is widely expressed throughout the developed brain, including the choroid plexus, cortex, hippocampus, thalamus, hypothalamus, cerebellum, and spinal cord (*Kar et al., 2006*), with higher expression in cortex, hippocampus, cerebellum, and certain brainstem nuclei (*Hawkes and Kar, 2004*). In addition, specific variants of the gene encoding CIM6P/IGF2R have been associated with high cognitive ability in a quantitative trait locus study (*Chorney et al., 1998*). CIM6P/IGF2R has also been implicated in neurodegenerative diseases, although the mechanisms underlying this effect remain to be established (*Wang et al., 2017*). In spite of this knowledge, relatively little is known about the function of the CIM6P/IGF2R in the mature central nervous system and its specific role in cognition.

Previous studies from our lab showed that IGF2 expressed in the hippocampus is required for long-term memory formation, and that administration of recombinant IGF2 either directly into the hippocampus or via systemic injection significantly enhances memories and prolongs their persistence (*Agis-Balboa et al., 2011*; *Chen et al., 2011*; *Stern et al., 2014*). Furthermore, IGF2 prevents aging-related memory loss in rats (*Steinmetz et al., 2016*), and reverses multiple deficits, including cognitive ones, in mouse models of Alzheimer's disease (*Mellott et al., 2014*; *Pascual-Lucas et al., 2014*). In addition, subcutaneous injections of IGF2 reversed most core deficits in a mouse model of autism spectrum disorders (*Steinmetz et al., 2018*). These studies also demonstrated that the effects of IGF2 in memory enhancement and recovery of function in disease models are dependent on CIM6P/IGF2R (*Chen et al., 2011*; *Steinmetz et al., 2018*; *Stern et al., 2014*).

Given these effects, in this study, we investigated the role of CIM6P/IGF2R in memory formation and elucidated the underlying mechanisms in adult rats and mice.

## Results

### CIM6P/IGF2R is highly expressed in neurons of all hippocampal subregions

Given our interest in investigating the role of CIM6P/IGF2R in hippocampus-dependent memories, we first determined the cellular localization of CIM6P/IGF2R in the rat hippocampus. To this end, we co-immunostained coronal brain sections with antibodies specific for CIM6P/IGF2R and markers selective for neurons, astrocytes, or microglia. The anti-CIM6P/IGF2R antibody was validated by a gene knockdown approach, as described below.

As depicted in *Figure 1a*, CIM6P/IGF2R immunostaining co-localized with the neuronal marker MAP2, but not with the astrocytic marker GFAP or the microglial marker Iba1, indicating that the receptor was mainly expressed in neurons. The expression was found throughout all hippocampal subregions, CA1, CA2, CA3 and dentate gyrus (DG) and also largely co-localized with calcium/calmodulin-dependent kinase II α (CaMKIIα; *Figure 1—figure supplement 1*), a marker of excitatory neurons (*Liu and Murray, 2012*). In neurons, the highest levels of CIM6P/IGF2R staining were concentrated in the cell body and particularly in the soma and proximal dendrites.

Next, we investigated whether learning regulates the expression levels of the CIM6P/IGF2R. Using the contextual fear conditioning-based inhibitory avoidance (IA) task in rats, which models aversive hippocampus-dependent episodic memories, we found that levels of neither *Igf2r* mRNA, measured with quantitative polymerase chain reaction (qPCR), nor CIM6P/IGF2R protein, measured by western blot analyses, changed 1 hour (hr) after training, relative to those of untrained controls (*Figure 1b and c*). In agreement with the immunofluorescence data showing the highest expression in the perinuclear area, western blot analyses comparing total with synaptoneurosomal extracts revealed that levels of CIM6P/IGF2R are significantly higher in the total protein homogenate (*Figure 1c*). Again, the level of CIM6P/IGF2R was unchanged 1 hr after training in both fractions. A more extended time course of CIM6P/IGF2R protein levels following IA training (30 min, 2 days, 1 week, and 2 weeks) measured with western blot analysis, also revealed no changes compared to untrained controls (*Figure 1d*). The hippocampal protein extracts were validated by assessing the

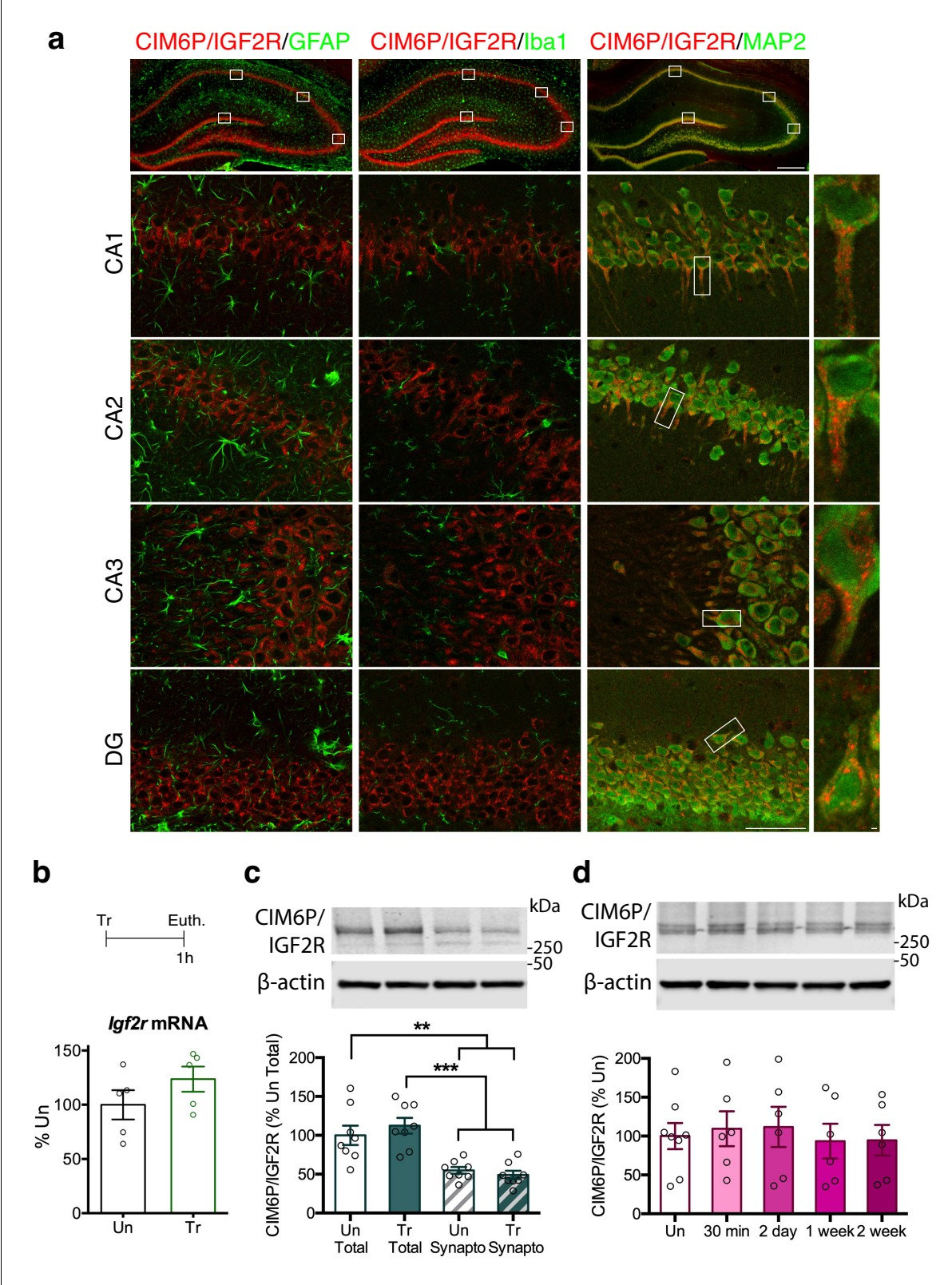

**Figure 1.** CIM6P/IGF2R is expressed in rat hippocampal neurons and mostly localizes to the somatic compartment. (**a**) Immunofluorescence co-staining of CIM6P/IGF2R and GFAP, Iba1, or MAP2. Upper panels: representative composite tile scans of whole hippocampus (scale bar, 500 μm). Lower panels: CA1, CA2, CA3, and DG (scale bar, 50 μm). Far right panels: zoomed images showing co-localization of MAP2 with CIM6P/IGF2R (scale bar, 1 μm). (**b**) Rats were trained on IA (Tr) or remained in their home cages (untrained, Un) and euthanized 1 hr after training. *Igf2r* mRNA levels (n = 5, two

*Figure 1 continued on next page*

*Figure 1 continued*

independent experiments). (**c**) Western blot analyses comparing total and synaptoneurosomal extracts (n = 8, two independent experiments). (**d**) Total extracts from rats euthanized at various time points after training (30 min, 2 days, 1 week, and 2 weeks) (n = 6–8, four independent experiments). Two-tailed Student t-test or one-way ANOVA followed by Tukey's *post-hoc* tests. **p<0.01 and ***p<0.001; see Source data one for detailed statistical information.

The online version of this article includes the following figure supplement(s) for figure 1:

**Figure supplement 1.** CIM6P/IGF2R is expressed in CaMKIIα neurons of rat hippocampus.

**Figure supplement 2.** Time course of Egr1 protein induction following IA training in rats.

rapid and transient training-dependent induction of the immediate early gene (IEG) Egr1 with western blot analysis (*Lonergan et al., 2010*; *Veyrac et al., 2014*). As expected, Egr1 protein level was significantly induced at 30 min after training and returned to baseline at the later timepoints (*Figure 1—figure supplement 2*).

## Hippocampal CIM6P/IGF2R is rapidly recruited by learning to form long-term memory

Using a specific CIM6P/IGF2R-blocking antibody (*Chen et al., 2011*; *Martin-Montañez et al., 2014*), we determined the temporal window during which CIM6P/IGF2R is functionally required in the dorsal hippocampus of rats following IA learning.

First, we reproduced the findings by *Chen et al. (2011)* showing that two injections of anti-CIM6P/IGF2R (5 ng) into dorsal hippocampus, one immediately after IA training and a second 8 hr later, impaired memory retention 1 day after training relative to control injections of IgG (5 ng) (*Figure 2a*). We then tested additional timepoints for memory retention and found that the effect of blocking CIM6P/IGF2R with the two injections was long-lasting: memory impairment persisted at 1 week, and memory was not re-instated after exposure to a reminder shock (RS) given in a distinct context. Again, consistent with the results of *Chen et al. (2011)*, we found that a single bilateral injection given either immediately after training, or 8 hr later was not able to impact memory retention (*Figure 2b and c*), indicating that the critical function of CIM6P/IGF2R in memory formation extends for a temporal window of several hours after training.

To determine whether single injection lacked an effect due to insufficient blockade of the receptor, we assessed a higher dose of anti-CIM6P/IGF2R. A single bilateral injection of a 10-fold higher dose of anti-CIM6P/IGF2R, 50 ng, given immediately after IA training or 8 hr later, again had no effect on memory retention (*Figure 2d and e*), leading us to conclude that the prolonged action of CIM6P/IGF2R is necessary for several hours after IA training in order for long-term memory to be formed.

To better frame the temporal window of CIM6P/IGF2R functional requirement and establish whether the receptor is needed for the learning (encoding) phase, we bilaterally injected anti-CIM6P/IGF2R 15 min before training, and then tested memory retention 1 day and 1 week later. Memory was completely disrupted by the pre-training injection of anti-CIM6P/IGF2R (*Figure 2f*), leading us to conclude that learning rapidly recruits CIM6P/IGF2R for memory formation, and that, once engaged by learning, the receptor remains functionally involved for several hours.

## Hippocampal CIM6P/IGF2R is required for memory consolidation but not for learning, memory retrieval, or reconsolidation

To further confirm that the receptor plays a fundamental role in memory formation, we next examined the effect of blocking CIM6P/IGF2R in the hippocampus before training by bilaterally injecting anti-CIM6P/IGF2R prior to a stronger IA training protocol (0.9 mA footshock, as opposed to 0.6 mA). Memory was also completely disrupted by this regimen (*Figure 3a*), confirming that CIM6P/IGF2R is recruited by learning and essential for long-term memory formation. However, using independent cohorts of rats, in order to avoid confounds of multiple testing, we found that pre-training injection of anti-CIM6P/IGF2R did not impact learning, as memory retention was intact when tested 5 min after training (*Figure 3b*). Memory impairment was however already significant at 1 hr after training (*Figure 3c*), indicating that CIM6P/IGF2R is controlling a rapid post-training function that is key for memory consolidation.

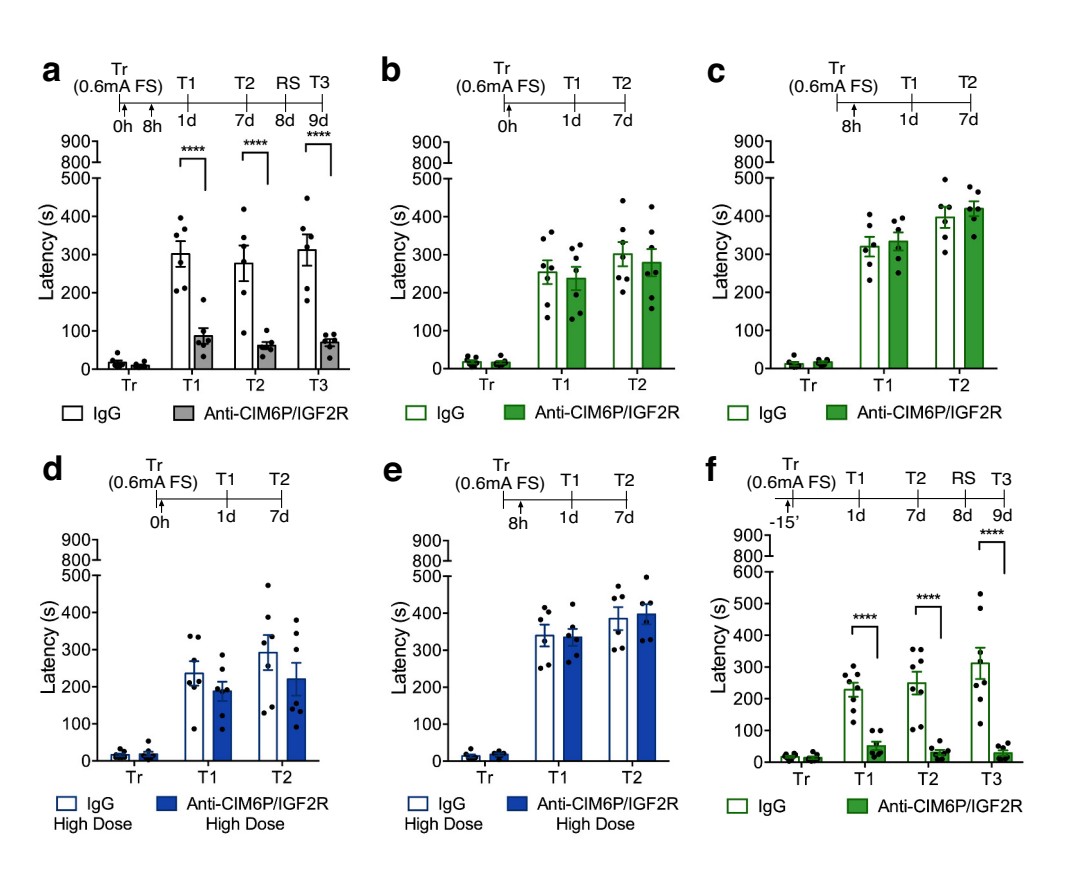

**Figure 2.** In rats, CIM6P/IGF2R is rapidly recruited by learning and required for memory consolidation within a limited temporal window. Experimental timelines are shown above graphs. Rats were injected with IgG or anti-CIM6P/IGF2R antibody before or after IA training (↑). IA acquisition (Tr) and memory retention are expressed as mean latency ± SEM (in seconds). (a) The effect of two injections of either IgG control or anti-CIM6P/IGF2R, given immediately and 8 hr after training (Tr), on IA memory tested 1 day (1 d) and 1 week after training, as well as after a reminder shock (RS; n = 6, two independent experiments). (b and c) The effect of a single injection of anti-CIM6P/IGF2R, given immediately or 8 hr after training, on memory tested 1 day and 1 week after training (n = 6–7, two independent experiments). (d and e) The effect of a single 10-fold higher dose of IgG or anti-CIM6P/IGF2R, given at the same time points, on memory tested 1 day and 1 week later (n = 6–7, two independent experiments). (f) The effect of a single injection, given 15 min before training, on memory tested 1 day and 1 week after training, as well as after a reminder shock (n = 7–8, two independent experiments). Two-way repeated measures ANOVA followed by Sidak's *post*-hoc tests. ****p<0.0001; see Source data one for detailed statistical information.

To determine whether the effect of anti-CIM6P/IGF2R injections was on memory and not due to other behavioral responses, another cohort of rats was used to assess locomotion and anxiety responses using the open field task. Specifically, we administered a bilateral anti-CIM6P/IGF2R injection 15 min prior to IA training, and then tested the injected rats in the open field 1 day later (*Figure 3d*). Both their locomotion (as measured by total distance travelled and average velocity) or anxiety measures (time spent in the center of the arena) were indistinguishable from those of IgG-injected controls. Moreover, anti-CIM6P/IGF2R injection had no effect on memory retrieval: a bilateral injection given 15 min before test at 1 day after training had no effect on memory retention in the 1 day or 1 week test (*Figure 3e*). The lack of an effect on the 7 day test also revealed that CIM6P/IGF2R is dispensable for memory reconsolidation, the process of fragility and re-stabilization produced by retrieval (*Alberini, 2011*).

Together, these results indicate that hippocampal CIM6P/IGF2R is essential for memory consolidation, but not for learning, memory retrieval, or reconsolidation.

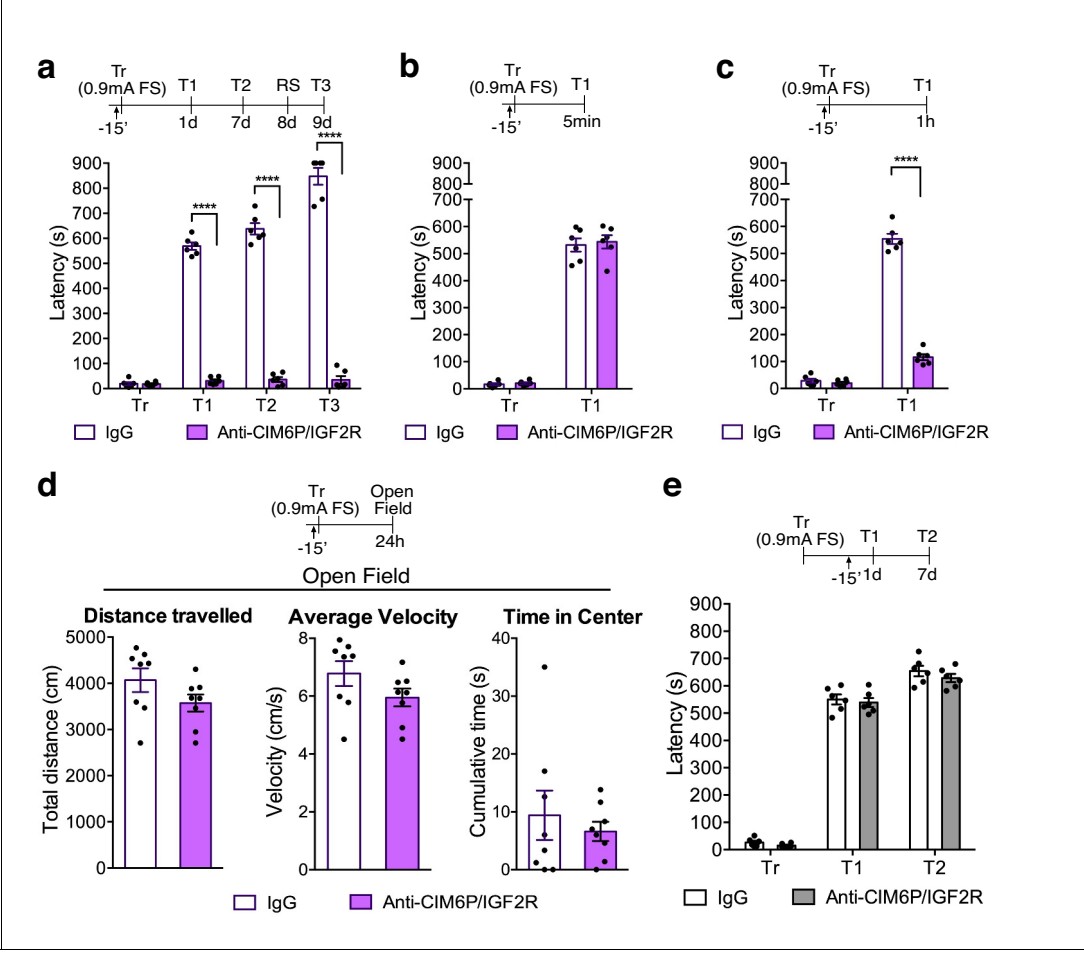

**Figure 3.** In rats, CIM6P/IGF2R is required for memory consolidation, but not learning or memory retrieval. Experimental timelines are shown above graphs. Rats were injected with IgG or anti-CIM6P/IGF2R antibody 15 min before IA training or testing (↑). Inhibitory avoidance training and memory retention are expressed as mean latency ± SEM (in seconds). (a) The effect of a single injection of anti-CIM6P/IGF2R on memory retention in animals trained (Tr) with a stronger, (0.9 mA) shock intensity (n = 6, two independent experiments). (b) The effect of anti-CIM6P/IGF2R on memory tested 5 min after training (n = 6, two independent experiments), or 1 hr after training (c) n = 6, two independent experiments). (d) Open field test was conducted 24 hr after IA training; total distance travelled, average travel velocity, and cumulative time spent in the center of the arena were recorded (n = 8, two independent experiments). (e) The effect of anti-CIM6P/IGF2R on memory retrieval: anti-CIM6P/IGF2R was injected 15 min before memory test given 1 day after IA training (n = 6, two independent experiments). Two-way repeated measures ANOVA followed by Sidak's *post hoc* tests. ****p<0.0001; see Source data one for detailed statistical information.

## Neuronal CIM6P/IGF2R is required for the formation of hippocampus-dependent memories

Given that CIM6P/IGF2R was mostly expressed by neurons in the hippocampus, we examined the effect of neuronal knockdown of CIM6P/IGF2R on memory processes using homozygous *Igf2r*-floxed mice. These experiments also allowed us to determine whether the function of CIM6P/IGF2R in long-term memory formation is conserved across different species. In these transgenic mice, exon 10 of the *Igf2r* gene is flanked by *loxP* sites, ready for excision and subsequent knockdown upon expression of Cre recombinase (*Wylie et al., 2003*). AAV-hSyn-Cre-GFP (Cre) or control AAV-hSyn-GFP (GFP) virus was bilaterally injected into dorsal hippocampi of *Igf2r*-floxed mice, and the viruses were allowed to express for 2 weeks before behavioral assessment.

The GFP- or Cre-injected mice were tested in a sequence of behavioral tasks (*Figure 4a*). First, the mice underwent open field test, which also served to habituate them to the arena, and 1 day later they were assessed in novel object location (nOL), a spatial memory task. After 1 week, the mice underwent Pavlovian fear conditioning, a task that allows for within-subject assessment of

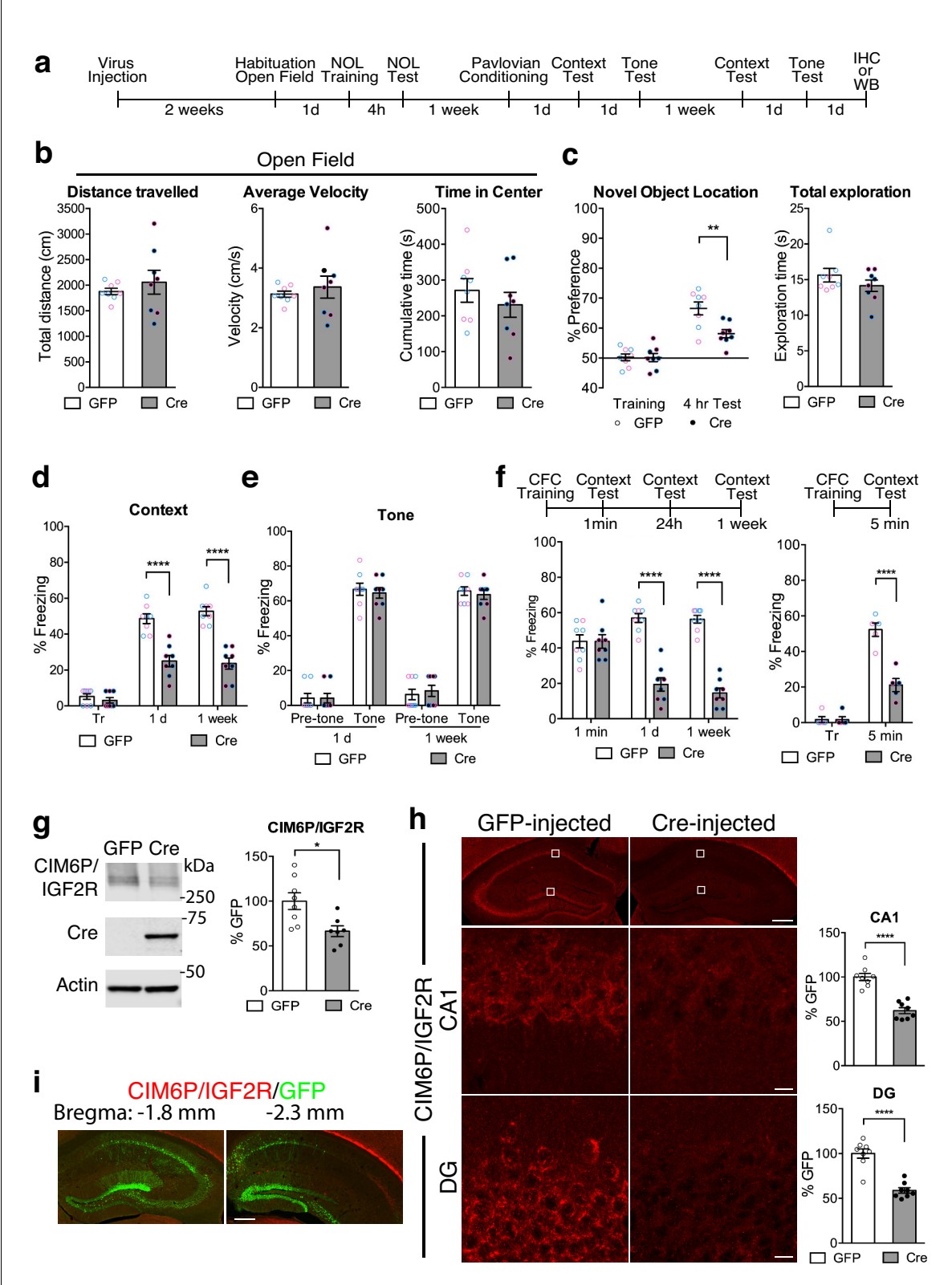

**Figure 4.** Hippocampal neuronal knockdown of CIM6P/IGF2R in mice selectively impairs hippocampus-dependent memories. (a) Experimental timeline for b–e and g–i. Data are expressed as mean ± SEM, with blue datapoints representing male mice, and pink datapoints representing female mice. *Igf2r*-floxed mice received bilateral hippocampal injections of AAV-hSyn-Cre-eGFP (Cre) or control AAV-hSyn-eGFP (GFP), 2 weeks prior to behavioral experiments. (b) Open field of GFP- and Cre-injected mice measured total distance travelled, average travel velocity, and cumulative time spent in the

*Figure 4 continued on next page*

*Figure 4 continued*

center of the area (n = 8, four independent experiments). (c) Novel objection location memory and total exploration time, tested 4 hr after training, (n = 8, four independent experiments). (d) Percent (%) of time spent freezing in mice tested in the training context 1 day (1 d) and 1 week after Pavlovian conditioning (n = 8, four independent experiments). (e) Percent (%) of time spent freezing to a new context prior to onset of tone, and during the tone (n = 8, four independent experiments). (f) Contextual fear conditioning expressed as percent (%) of time spent freezing. Left: % of time spent freezing in mice tested 1 min, 1 day, and 1 week after training (n = 8, four independent experiments). Right: % of time spent freezing in mice tested 5 min after training. (g) Representative western blot and quantification obtained from dorsal hippocampi homogenates from Cre- or GFP-injected mice stained for CIM6P/IGF2R, Cre recombinase (Cre), and actin. Actin-normalized values were expressed as mean percentage ± SEM (n = 7–8, four independent experiments). (h) Upper panels: representative dorsal hippocampus composite tile scans in GFP-injected or Cre-injected mice immunostained for CIM6P/IGF2R (scale bar, 500 µm). Middle and lower panels: CA1 and DG (scale bar, 10 µm) are shown. Bar graphs on the right report immunostaining intensity quantifications for each sub-region (n = 8, four independent experiments). (i) Representative images of GFP- or Cre-injected mice, composite tile scans of dorsal hippocampus double staining with anti-GFP and anti-CIM6P/IGF2R antibodies (scale bar 500 µm). Two-tailed Student's t-test or two-way repeated measures ANOVA followed by Sidak's *post*-hoc tests. *p<0.05, ****p<0.0001; see Source data one for detailed statistical information.

hippocampus-dependent (contextual fear conditioning) *vs.* hippocampus-independent (cued fear conditioning) fear memory (*Kim and Fanselow, 1992*; *Phillips and LeDoux, 1992*).

In the open field paradigm, both GFP- and Cre-injected mice had similar distance travelled and average velocity, indicating no change in locomotion, and spent a comparable amount in the center of the arena, indicating similar levels of anxiety response (*Figure 4b*).

However, compared to GFP-injected controls, Cre-injected mice exhibited impaired memory for nOL when tested 4 hr after training, despite having similar total exploration time (*Figure 4c*). Cre-injected mice also had significantly impaired contextual fear memory (*Figure 4d*) but intact tone (cued) conditioning when tested 1 day later (*Figure 4e*). When mice were then tested again a week later for contextual conditioning, the contextual fear memory of the Cre-injected mice remained impaired. The fact that Cre-injected mice, despite their deficit in contextual freezing, significantly froze in response to the tone indicated that Cre-mediated knockdown of CIM6P/IGF2R did not interfere with cued-conditioning or the ability to freeze. These data also confirmed that the knockdown of CIM6P/IGF2R selectively targeted memories requiring the hippocampus, the region targeted by viral injections.

To determine whether the function of the CIM6P/IGF2R in memory consolidation is also rapidly engaged in mouse, like in rat memories, we subjected two separate cohorts of mice to contextual fear conditioning. In the first cohort, tested 1 min, 1 day, and 1 week after training, both GFP- and Cre-injected mice exhibited robust freezing at 1 min, but Cre-injected mice had significant memory loss at 1 day and 1 week after training (*Figure 4f*). To better frame the temporal window of the CIM6P/IGF2R engagement, the second cohort was tested 5 min after training. As shown in *Figure 4f*, Cre-injected mice already had significant memory impairment at this very early timepoint.

Thus, in agreement with the experiments in rats, these data indicated that when CIM6P/IGF2R is knocked down from dorsal hippocampal neurons of mice, their ability to learn hippocampus-dependent tasks is intact, but memories fail to undergo consolidation. Additionally, our findings confirmed that the functional engagement of CIM6P/IGF2R following learning is very fast, and is required for rapid changes critical for memory consolidation.

Western blot (*Figure 4g*) and immunofluorescence analyses (*Figure 4h*) showed that Cre-recombinase expression via viral injection led to 40–50% CIM6P/IGF2R knockdown relative to GFP-injected controls. Immunofluorescence analyses confirmed that CIM6P/IGF2R was knocked down selectively where the GFP reporter was expressed, proving the specificity of both viral-mediated knockdown and antibody reactivity (*Figure 4i*).

## CIM6P/IGF2R controls the training-induced upregulation of IEG proteins, but not mRNAs

Immediate early genes (IEGs) such as Arc, Fos, and Egr1 (also known as Zif268) are rapidly induced at both the transcriptional and translational levels in response to learning, and their induction is required for memory formation (*Gallo et al., 2018*; *Minatohara et al., 2016*; *Tischmeyer and Grimm, 1999*). Given the essential role of CIM6P/IGF2R on memory consolidation, we asked whether blocking the receptor would affect the induction of IEGs at the mRNA and protein levels.

To achieve precise temporal control, we injected anti-CIM6P/IGF2R or IgG control into hippocampi of rats 15 min prior to IA training. Hippocampi from half of the rats were dissected and flash-frozen 1 hr after training to examine mRNA levels, whereas the other rats were perfused, and their coronal sections subjected to immunostaining. As expected, qPCR revealed that *Arc*, *Egr1*, and *c-Fos* mRNA levels were all significantly elevated following training in IgG-injected rats relative to untrained controls. Notably, equivalent inductions were also seen in trained rats that received bilateral hippocampal injection of anti-CIM6P/IGF2R (*Figure 5a*), indicating that CIM6P/IGF2R is not required for the mRNA induction of these three IEGs. By contrast, immunostaining revealed that the intensity of Arc, Egr1, and c-Fos proteins was increased in response to training, and completely abolished by anti-CIM6P/IGF2R (*Figure 5b,c,d*). Specifically, these immunohistochemical analyses quantified two measures in both hippocampal subregions CA1 and DG: i) the total immunostaining intensity and ii) the number of cells positive for each IEG. Both parameters were normalized for the number of total cells assessed with DAPI staining (see Methods).

We found that training significantly increased the intensity of all three IEGs in both CA1 and DG (*Figure 5—figure supplement 1*). Furthermore, training significantly increased the number of Arc-positive cells in both CA1 and DG and the number Egr1- and c-Fos-positive cells in DG. In contrast, although their intensity significantly increased with training, the number of Egr1- or c-Fos-positive cells did not change in CA1. In sum, training led to a significant upregulation of IEG protein levels in both CA1 and DG hippocampal subregions, but with distinctive IEG expression modalities in CA1 where either the number of cells expressing the IEG increased (i.e. Arc) or cells already expressing the IEG (Erg1 and c-Fos) increased their levels. In DG, training led to a significant increase in the number of positive cells for all three IEGs. All these changes required CIM6P/IGF2R, as they were abolished by anti-CIM6P/IGF2R.

We concluded that the training-induced increase in Arc, c-Fos and Egr1 proteins, but not the corresponding mRNAs, requires CIM6P/IGF2R.

## CIM6P/IGF2R is required for the training-induced increase in *de novo* protein synthesis

*De novo* protein synthesis is rapidly induced in response to learning and is required for long-term memory formation (*Costa-Mattioli et al., 2009*; *Richter and Klann, 2009*; *Sutton and Schuman, 2006*; *Wang et al., 2009*). Given that the training-induced increase in proteins, but not mRNAs, of all three IEGs was affected by CIM6P/IGF2R, we asked whether the receptor controls the learning-dependent increase in *de novo* protein synthesis in general. To this end, we employed *in vivo* SUrface SEnsing of Translation (SUnSET), which measures incorporation of puromycin into elongating peptide chains (*Schmidt et al., 2009*), thus assessing active translation (*Descalzi et al., 2019*; *Goodman et al., 2011*; *Santini et al., 2013*). Puromycin was co-injected with anti-CIM6P/IGF2R or IgG bilaterally into rat hippocampi 15 min prior to IA training or in untrained control rats. All animals were perfused 2 hr after injection, and coronal brain sections were subjected to immunostaining to measure the puromycin signal. As expected, training led to significant induction of puromycin incorporation throughout the dorsal hippocampus, which was especially prominent in neuronal populations, but this induction was completely abolished by anti-CIM6P/IGF2R (*Figure 6a*). Together, these data indicate that CIM6P/IGF2R controls the training-dependent increase in *de novo* protein synthesis.

## Neuronal CIM6P/IGF2R is a target mechanism for memory enhancement

Previous studies showed that both hippocampal and systemic administration of IGF2 significantly enhances memory retention and persistence (*Chen et al., 2011*; *Stern et al., 2014*), and this effect requires hippocampal CIM6P/IGF2R (*Chen et al., 2011*; *Stern et al., 2014*). To determine whether the memory-enhancing effect of IGF2 is a function of the receptor, rather than IGF2 *per se*, we investigated a distinct CIM6P/IGF2R ligand, M6P. Vehicle or varying doses of M6P (0.005, 5, 25 and 150 mM) were injected bilaterally into the rat dorsal hippocampus immediately after IA training to obtain a dose-response curve. We found that 5 mM and 25 mM, but not the other doses, significantly enhanced memory retention (*Figure 7a*). The effect was long-lasting, and the memory enhancement

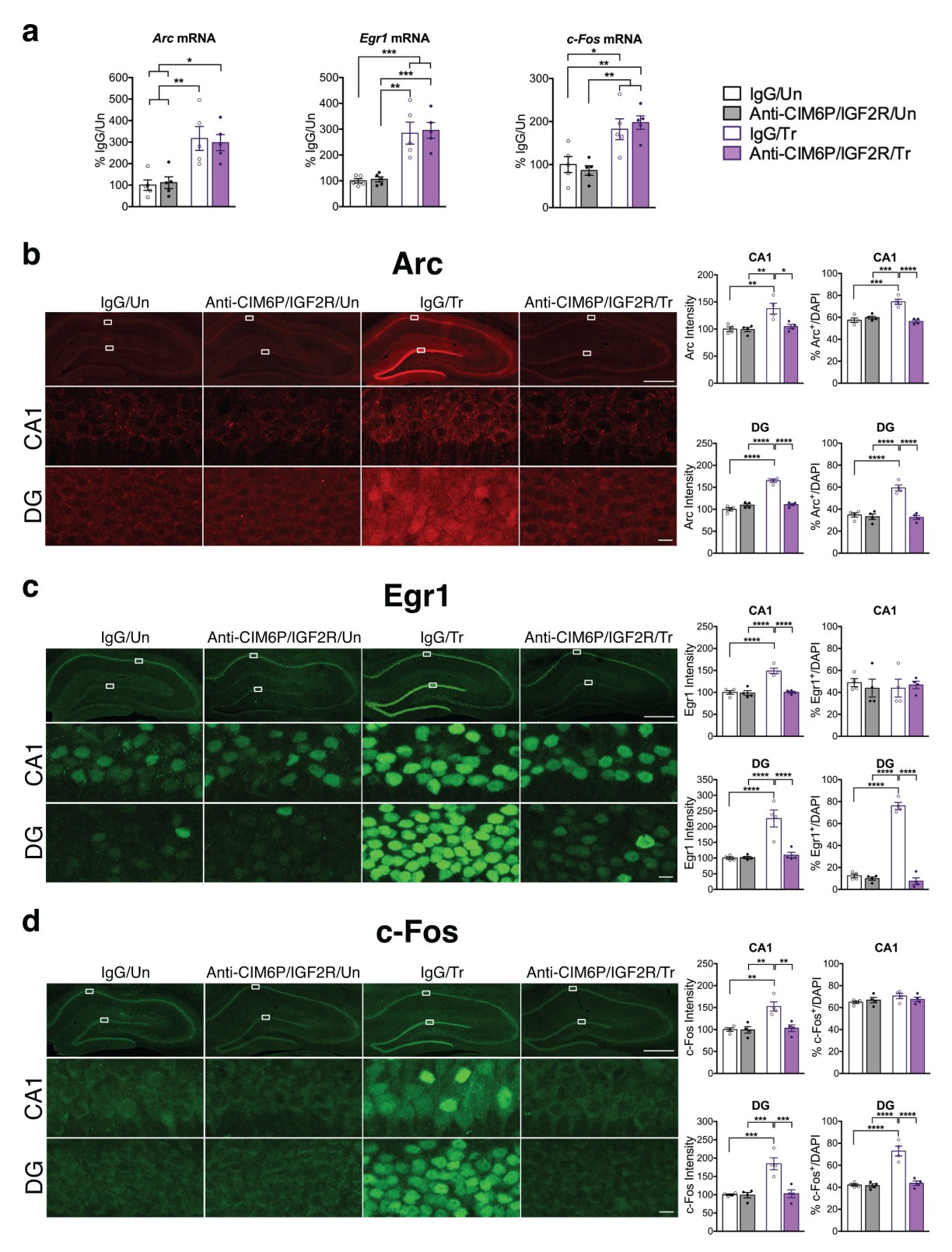

**Figure 5.** CIM6P/IGF2R is required for learning-induced IEG protein induction in rats. Rats were injected with IgG or anti-CIM6P/IGF2R, and then were either trained on IA 15 min later (Tr) or remained Untrained (Un). To examine mRNA levels, rats were euthanized 1 hr after training and hippocampi flash-frozen. (a) qPCR of *Arc, Egr1, and c-Fos* performed on dorsal hippocampal extracts obtained from rats injected with either IgG or anti-CIM6P/IGF2R and euthanized 1 hr after IA training (n = 5, two independent experiments). (b–d) Rats underwent IA training and were perfused 1 hr later;

*Figure 5 continued on next page*

*Figure 5 continued*

coronal brain sections were immunostained for Arc (**b**), Egr1 (**c**), and c-Fos (**d**). Upper panels: representative composite tile scan of dorsal hippocampus (scale bar, 500 μm). Middle and lower panels: CA1 and DG (scale bar, 10 μm). Bar graphs shown at right show quantifications for normalized intensity and percentage of cells positive for IEGs, for each sub-region (n = 4, two independent experiments). Two-way ANOVA followed by Tukey's *post hoc* tests. *p<0.05, **p<0.01, ***p<0.001, ****p<0.0001; see Source data one for detailed statistical information.

The online version of this article includes the following figure supplement(s) for figure 5:

**Figure supplement 1.** Total number of cells assessed by DAPI staining in the rat dorsal hippocampus fields used in the IEGs immunofluorescent analyses.

persisted 1 week after training. These data indicated that CIM6P/IGF2R ligands, as opposed to IGF2, have memory-enhancing effects.

We then determined whether CIM6P/IGF2R expressed by hippocampal neurons is a target mechanism for IGF2- and M6P-induced memory enhancement. Again, we bilaterally injected of GFP- or Cre-expressing viruses into *Igf2r*-floxed mice, and 2 weeks later injected them systemically (s.c.) with vehicle, IGF2, or M6P before subjecting them to contextual fear conditioning training. The GFP-injected mice that received IGF2 or M6P exhibited significant memory enhancement 1 day after training relative to those that received vehicle, and the enhancement persisted 1 week later (*Figure 7b*). The Cre-injected mice that received vehicle, as expected, exhibited significant memory impairment relative to the GFP-injected mice that received vehicle, and IGF2 or M6P administration had no effect on their memory impairment.

Collectively, these results showed that both ligands of CIM6P/IGF2R, IGF2 and M6P, significantly enhance memory retention and persistence by acting through CIM6P/IGF2R expressed by neurons in the hippocampus. These findings identify this receptor as a mechanism that can promote memory enhancement.

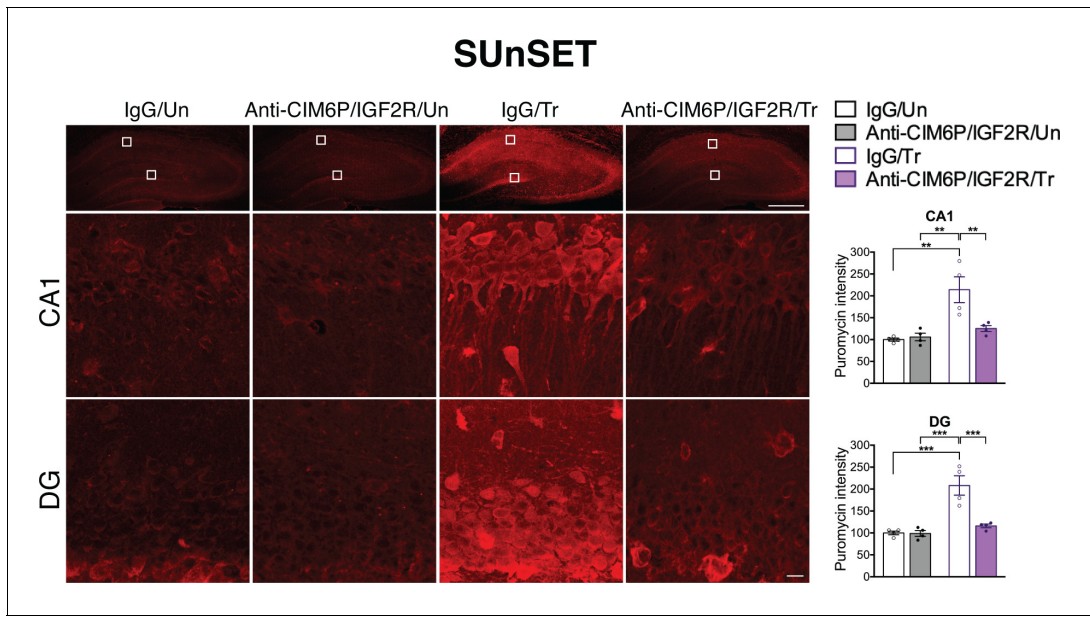

**Figure 6.** CIM6P/IGF2R is required for learning-induced *de novo* protein synthesis in rats. SUnSET was employed to quantify *de novo* protein synthesis in the rat hippocampus. Puromycin was co-injected with IgG or anti-CIM6P/IGF2R prior to training, and rats were perfused 2 hr later. Upper panels: representative images of anti-puromycin immunostaining, composite tile scans of whole hippocampus (scale bar, 500 μm). Middle and lower panels: CA1 and DG (scale bar 10 μm). Bar graphs at right show immunostaining intensity quantifications for each sub-region (n = 4, two independent experiments). Two-way ANOVA followed by Tukey's *post hoc* tests. **p<0.01, ***p<0.001; see Source data one for detailed statistical information.

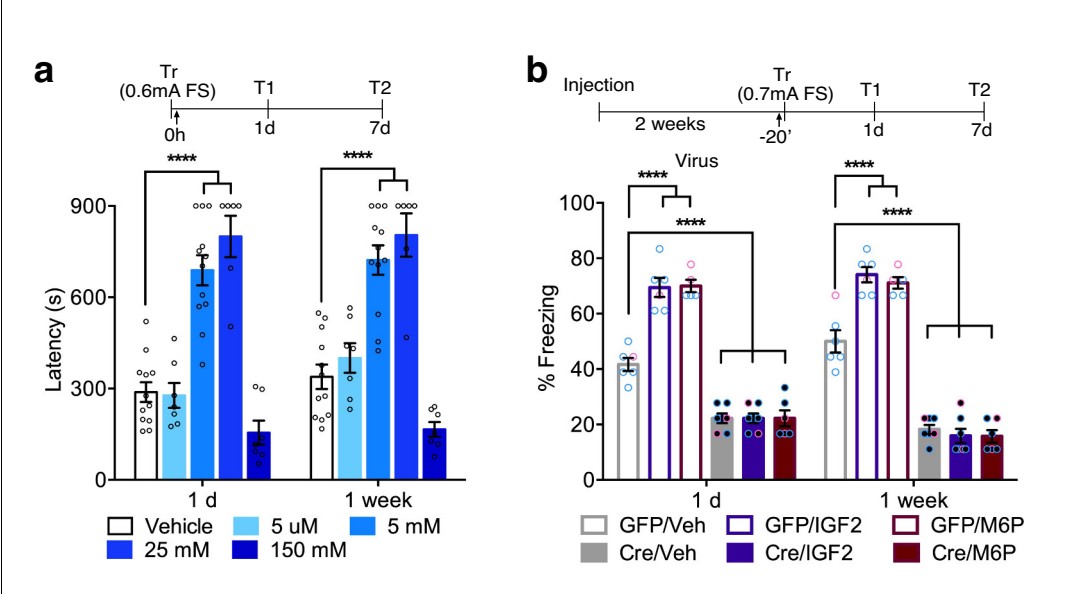

**Figure 7.** In rats and mice, M6P, like IGF2, enhances memories via CIM6P/IGF2R in hippocampal neurons. Experimental timelines are shown above graphs. (a) Rats were injected bilaterally into the dorsal hippocampus with vehicle or the indicated concentrations of mannose-6-phosphate (M6P) immediately after IA training (↑), and memory was tested 1 day and 1 week later (n = 6–12, five independent experiments). (b) Vehicle, IGF2 (30 µg/kg), or M6P (850 µg/kg) was administered systemically (s.c., ↑) 20 min before contextual fear conditioning training to GFP- or Cre-injected mice. Percent (%) of time spent freezing were measured 1 day or 1 week after training (n = 5–7, five independent experiments). Two-way repeated measures ANOVA followed by Sidak's *post hoc* tests. ****p<0.0001; see Source data one for detailed statistical information.

## Discussion

Our results demonstrate that CIM6P/IGF2R expressed by hippocampal neurons is required for the consolidation, but not learning, retrieval or reconsolidation of hippocampus-dependent memories in rodents. We also found that learning rapidly engages CIM6P/IGF2R to increase protein levels in response to learning, including the IEGs Arc, Egr1, and c-Fos. However, the receptor is dispensable for the learning-dependent increase in the mRNA levels of these IEGs. Finally, administration of the receptor ligand M6P, like IGF2, significantly enhances long-term memory, indicating that CIM6P/IGF2R represents a target mechanism for memory enhancement.

These data identified novel functional roles of the CIM6P/IGF2R, which are added to its previously known functions on the regulation of organ growth and development, tumor suppression, and its possible involvement in Alzheimer's disease (*El-Shewy and Luttrell, 2009*; *Wang et al., 2017*). Because CIM6P/IGF2R is abundantly expressed in the mature brain, particularly in neurons, it is important to understand its contributions to brain functions under normal conditions and in disease states.

Our acute inhibition of CIM6P/IGF2R at various timepoints before and after training revealed that the receptor in the dorsal hippocampus is not required for encoding, retrieval, or reconsolidation but is rapidly engaged by learning to allow the regulation of immediate changes required for memory consolidation. In fact, blockade of the receptor pre-training completely abolished memory and led to memory impairment very rapidly: at 1 hr after IA learning in rats, and 5 min after contextual fear conditioning in mice. These results indicated that the CIM6P/IGF2R is promptly recruited by training and regulates rapid mechanisms critical for memory consolidation. However, once engaged by learning, the function of CIM6P/IGF2R in memory consolidation continued for several hours: only a prolonged post-training blockade of the receptor by two injections, one immediately after training and another 8 hr later, produced memory impairment. The reason for this prolonged functional requirement of CIM6P/IGF2R following training remains to be fully understood, and future studies are needed to address this issue.

Rapid changes induced by learning that then persists for several hours or days as critical mechanisms of memory consolidation include *de novo* transcription and translation, hallmark molecular

signatures of long-term memory (*Alberini and Kandel, 2015*; *Costa-Mattioli et al., 2009*; *Santini et al., 2014*). Our data showed that CIM6P/IGF2R is necessary for the induction of *de novo* protein synthesis in response to learning, including the increase of the IEGs Arc, Egr1, and c-Fos proteins. In fact, receptor blockade abolished their induction specifically at the protein level, without affecting induction of the corresponding mRNAs. Thus, CIM6P/IGF2R is a critical upstream regulator of *de novo* protein synthesis increase evoked by learning.

These results provided key information on mechanisms downstream of the CIM6P/IGF2R; however, the specific mechanistic steps that link the receptor to *de novo* translation, and particularly translation of IEGs, remain to be understood. Because CIM6P/IGF2R has a small intracellular domain and does not have known direct signal transduction mechanisms (*El-Shewy and Luttrell, 2009*), there are no predictable hypotheses about which molecular mechanisms directly activate its downstream steps. We speculate that these mechanisms reside in domains that link of endosomal trafficking, lysosomal degradation and Golgi functions to learning-induced *de novo* translation, a complex area of investigation that shall be explored in future studies.

Previous studies have shown that multiple waves of regulated *de novo* protein synthesis are evoked by learning and required for memory consolidation (*Alberini, 2009*; *Barzilai et al., 1989*; *Bourtchouladze et al., 1998*; *Stork and Welzl, 1999*; *Taubenfeld et al., 2001*). More specifically, it has been found that IA training in rats very rapidly activates *de novo* translation in the hippocampus, and this translation is required for memory consolidation. This critical *de novo* protein synthesis continues, presumably through multiple waves, for more than 24 hr after training, returning to control conditions by 48 hr post-training (*Arguello et al., 2013*; *Bambah-Mukku et al., 2014*; *Chen et al., 2011*; *Garcia-Osta et al., 2006*; *Taubenfeld et al., 2001*). Our data showed that while a pre-training injection of receptor blocking antibody completely blocks long-term memory formation, a single injection given after training has no effect. This suggests that, once engaged, the receptor remains functionally critical for some time. In agreement with this explanation, two injections 8 hr apart given after training, but not single injections or even a single injection of 10x higher concentration of blocking antibody, were able to impair long-term memory. Together with our *in vivo* SUnSET data showing that the receptor controls *de novo* translation induced by learning, these results lead us to speculate that CIM6P/IGF2R is intimately coupled to the induction and persistence of experience-regulated *de novo* translation. If this were the case, we would expect that the inhibition of the receptor and of protein synthesis have similar temporal profiles of functional requirement. This is indeed the case, as, like with blockers of the receptor, a pre-training injection of the protein synthesis inhibitor anisomycin completely blocks long-term memory formation, whereas post-training injections have graded effects for the first 24 hr after training (*Bambah-Mukku et al., 2014*).

Nevertheless, the exact mechanisms by which CIM6P/IGF2R controls the increase in protein production remains to be determined, and ours are only speculative explanations. Importantly, our experiments could not dissect whether the decrease in puromycin incorporation upon CIM6P/IGF2R blockade was due to inhibition of mRNA translation or acceleration of protein degradation. In addition, a recent study reported that mRNA translation in axons of retinal ganglion cells occurs on ribosomes associated with late endosomes docked at mitochondria (*Cioni et al., 2019*). Given that CIM6P/IGF2R traffics cargo via endosomes, we speculate that the receptor may regulate endosomal availability and trafficking required for protein synthesis. Moreover, because CIM6P/IGF2R is crucial for maintaining lysosomal function (*Ludwig et al., 1996*; *Wang et al., 1994*), it may be involved in the regulation of protein degradation, which is activated in long-term plasticity and memory (*Bingol and Sheng, 2011*; *Fernández-Monreal et al., 2012*; *Goo et al., 2017*).

The rapid engagement of CIM6P/IGF2R implies that of one of its major ligands, IGF2, is rapidly recruited; however, previous studies showed that IGF2 is significantly upregulated by training not immediately but several hours (20 hr) later (*Chen et al., 2011*). It is possible that the rapid activation of the receptor occurs through other types of ligands, as the CIM6P/IGF2R can indeed bind a variety of molecules including mannosylated proteins such as lysosomal enzymes, TGF-β1 precursor, proliferin, plasminogen, and retinoic acid (*El-Shewy and Luttrell, 2009*).

It is also possible that IGF2 present at the basal level is released upon training, thus immediately recruiting the receptor, without changing its expression level for several hours. IGF2 upregulation at the level of transcription and translation may be needed for the persistence of the consolidation process. A similar type of regulation has been documented for the neurotrophin, brain derived neurothrophic factor (BDNF), which is immediately engaged but its requirement persists via an

autoregulatory positive feedback loop throughout the whole molecular consolidation temporal window (*Bambah-Mukku et al., 2014*).

Finally, the signaling capabilities of CIM6P/IGF2R remain controversial: some authors suggest that CIM6P/IGF2R directly or indirectly activates G-proteins (*El-Shewy et al., 2006*; *Hawkes et al., 2006*; *Ikezu et al., 1995*; *McKinnon et al., 2001*; *Nishimoto et al., 1989*; *Okamoto et al., 1990*), whereas others have presented opposing evidence (*Körner et al., 1995*; *Sakano et al., 1991*). Therefore, further studies are needed to elucidate whether signaling activated by CIM6P/IGF2R is involved in the regulation of *de novo* protein synthesis or degradation.

We also showed that expression of CIM6P/IGF2R in the hippocampus is expressed predominantly in neurons, as demonstrated by co-localization of CIM6P/IGF2R with MAP2 (a neuronal marker) but not GFAP (an astrocytic marker) or Iba1 (a microglial marker). These results are in agreement with several previous reports revealing relatively high [$^{125}$I]-IGF2 radio-labelled binding in the hippocampus (*Hawkes and Kar, 2003*; *Kar et al., 1993*; *Lesniak et al., 1988*; *Mendelsohn et al., 1988*; *Smith et al., 1988*), as well as other immunostaining studies with different anti-CIM6P/IGF2R antibodies (*Hawkes and Kar, 2003*; *Kar et al., 2006*; *Lesniak et al., 1988*; *Nolan et al., 1987*; *Zhou et al., 1995*).

Our data also indicated that training does not alter total levels of CIM6P/IGF2R mRNA or protein, leading us to conclude that although CIM6P/IGF2R is critical for memory consolidation, its total level does not change following learning. However, our experiments do not exclude the possibility that learning changes the intracellular distribution and trafficking of the CIM6P/IGF2R.

Cre-recombinase-mediated knockdown of neuronal CIM6P/IGF2R resulted in impairments of long-term, but not short-term, hippocampus-dependent memories without affecting non–hippocampus-dependent behaviors. This indicates that similar to rats, mice also require hippocampal CIM6P/IGF2R for memory consolidation, but not the ability to learn. The receptor specifically contributes to memory consolidation, but it is not required for other behavioral responses such as locomotion or anxiety responses.

Finally, our data also showed that the CIM6P/IGF2R can be targeted to enhance memory. Consistent with previous reports that CIM6P/IGF2R is required for the memory-enhancing effect of hippocampal or systemically administered IGF2 (*Chen et al., 2011*; *Stern et al., 2014*), we showed that a distinct ligand of the receptor, M6P, injected either in the hippocampus in rat or systemically in mouse, significantly increased memory strength and persistence. These effects were completely abolished in mice that had reduced receptor levels in their dorsal hippocampi, strengthening the conclusion that memory enhancement promoted by either ligand is indeed mediated by CIM6P/IGF2R in hippocampal neurons.

CIM6P/IGF2R has been implicated in several aspects of Alzheimer's disease (AD): levels of CIM6P/IGF2R decrease with the presence of APOE4 alleles (*Kar et al., 2006*), and overexpression of CIM6P/IGF2R increases amyloid precursor protein processing and amyloid-beta (Aβ) production (*Wang et al., 2015*), a hallmark alteration in AD. Furthermore, CIM6P/IGF2R co-localizes with subsets of Aβ-positive neuritic plaques (*Nixon, 2005*; *Wang et al., 2017*), and may also exert neuroprotective effects (*Martin-Montañez et al., 2014*; *Mellott et al., 2014*; *Mellott et al., 2017*). In light of our observations showing that neuronal CIM6P/IGF2R in the hippocampus plays a crucial role in memory consolidation and is a target mechanism for memory enhancement, this receptor may represent a key target for novel therapies for AD.

## Materials and methods

### Key resources table

| Reagent type (species) or resource | Designation | Source or reference | Identifiers | Additional information |
|---|---|---|---|---|
| Strain, strain background (*R. norvegicus*, male) | BluHsd:LE Long-Evans (blue spruce) | Envigo | RRID:RGD_5508398 | |

*Continued on next page*

*Continued*

| Reagent type (species) or resource | Designation | Source or reference | Identifiers | Additional information |
|---|---|---|---|---|
| Strain, strain background (*M. musculus*, male and female) | *Igf2r*-floxed mice | Dr. David Skaar (NC State University | MGI Cat# 3795370, RRID:MGI:3795370 | C57Bl/6J background, homozygotes used in experiments |
| Antibody | anti-Human IGF-II R (goat polyclonal) | R and D Systems | Cat# AF2447, RRID:AB_442153 | 5 ng/µL or 50 ng/µL |
| Antibody | anti-GFAP (chicken polyclonal) | Abcam | Cat# ab4674, RRID:AB_304558 | IF (1:5000) |
| Antibody | anti-Iba1 (rabbit polyclonal) | Wako | Cat# 019–19741, RRID:AB_839504 | IF (1:5000) |
| Antibody | anti-CaMKIIα (mouse monoclonal) | Millipore | Cat# 05–532, RRID:AB_309787 | IF (1:1000) |
| Antibody | anti-IGF2R (rabbit monoclonal) | Abcam | Cat# ab124767, RRID:AB_10974087 | IF (1:1000), WB (1:1000) |
| Antibody | anti-GFP (chicken polyclonal) | Aves Labs | Cat# GFP-1020, RRID:AB_10000240 | IF (1:1000) |
| Antibody | anti-Arc (rabbit polyclonal) | Synaptic systems | Cat# 156 003, RRID:AB_887694 | IF (1:2000) |
| Antibody | anti-Egr1 (rabbit monoclonal) | Cell Signaling Technology | Cat# 4153, RRID:AB_2097038 | IF (1:1000), WB (1:1000) |
| Antibody | anti-c-Fos (rabbit monoclonal) | Cell Signaling Technology | Cat# 2250, RRID:AB_2247211 | IF (1:500) |
| Antibody | anti-Cre (rabbit monoclonal) | Cell Signaling Technology | Cat# 15036, RRID:AB_2798694 | WB (1:1000) |
| Antibody | anti-β-actin (mouse monoclonal) | Santa Cruz Biotechnology | Cat# sc-47778 HRP, RRID:AB_2714189 | WB (1:10000) |
| Sequence-based reagent | *Igf2r* forward | NM_012756.2 | Primers | TTGCCCTCCAGAAACGGAAG |
| Sequence-based reagent | *Igf2r* reverse | NM_012756.2 | Primers | TACACCACAGTTTCGCTCGT |
| Sequence-based reagent | *Arc* forward | NM_019361.1 | Primers | CCCTGCAGCCCAAGTTCAAG |
| Sequence-based reagent | *Arc* reverse | NM_019361.1 | Primers | GAAGGCTCAGCTGCCTGCTC |
| Sequence-based reagent | *c-Fos* forward | NM_022197.2 | Primers | CCCTGCAGCCCAAGTTCAAG |
| Sequence-based reagent | *c-Fos* reverse | NM_022197.2 | Primers | GAAGGCTCAGCTGCCTGCTC |
| Sequence-based reagent | *Egr1* forward | NM_012551.2 | Primers | ACCTACCAGTCCCAACTCATC |
| Sequence-based reagent | *Egr1* reverse | NM_012551.2 | Primers | GACTCAACAGGGCAAGCATAC |
| Sequence-based reagent | *Gapdh* forward | NM_017008.4 | Primers | GAACATCATCCCTGCATCCA |
| Sequence-based reagent | *Gapdh* reverse | NM_017008.4 | Primers | CCAGTGAGCTTCCCGTTCA |
| Peptide, recombinant protein | Recombinant mouse IGF-II | R and D Systems | Cat# 792 MG | |
| Commercial assay or kit | RNeasy Plus Universal Mini Kit | Qiagen | Cat# 73404 | |
| Commercial assay or kit | QuantiTect Reverse Transcription Kit | Qiagen | Cat# 205311 | |

*Continued on next page*

*Continued*

| Reagent type (species) or resource | Designation | Source or reference | Identifiers | Additional information |
|---|---|---|---|---|
| Commercial assay or kit | iQ SYBR Green Supermix | Bio-Rad | Cat# 107–8882 | |
| Software, algorithm | ImageJ | National Institutes of Health | RRID:SCR_003070 | |
| Software, algorithm | Leica Application Suite X | Leica | RRID:SCR_013673 | |

## Animals

Adult male Long-Evans rats weighing between 200 and 250 g at the beginning of experiments were used in this study. Rats were housed individually after cannulation surgeries. Homozygous male and female *Igf2r*-floxed mice (a generous gift from Dr. David Skaar, NC State University) were bred in house. Mice, which were 8 weeks old at the beginning of experiments were group housed. All animals were maintained on a 12 hr light/dark cycle, and all experiments were performed during the light cycle. Animals were provided with *ad libitum* access to food and water and were handled for 3 min per day for 5 days prior to any behavioral procedure. Animals were randomly assigned to treatment or behavioral groups for all experiments. All protocols complied with the National Institutes of Health Guide for the Care and Use of Laboratory Animals and were approved by the Institutional Animal Care and Use Committee at New York University.

## Cannula implants and hippocampal injections in rats

Cannula implants targeting the dorsal hippocampus were performed as described previously (*Chen et al., 2011*; *Ye et al., 2017*). Rats were anesthetized with ketamine (75 mg/kg, intraperitoneally [i.p.]) and xylazine (10 mg/kg, i.p.), and stainless-steel guide cannulae (22-gauge) were implanted to bilaterally target the dorsal hippocampus (dHC, 4.0 mm posterior to bregma, 2.6 mm lateral to midline, 2 mm ventral to skull surface) using a stereotaxic apparatus. Rats were administered meloxicam (3 mg/kg, subcutaneously, once pre-surgery), and allowed to recover from surgery in their home cage for at least 7 days before undergoing behavioral experiments. At the indicated time points before or after training or retrieval, rats received bilateral injections of compounds as specified. All injections were delivered in a volume of 1.0 μL over 3 min using a 28-gauge needle (extending 1.5 mm beyond the guide cannula) attached to polyethylene tubing (PE50) connected to 10 μL Hamilton syringes controlled by a micro-infusion pump. Infusions were delivered at a rate of 0.33 μL/min, and the injection needle was left in place for 2 min after the injection to allow complete dispersion of the solution. Cannula placement was verified at the end of the behavioral experiments. To this end, brains were snap-frozen, and 40 μm coronal sections were cut through the hippocampus, and examined under a light microscope. Twelve (out of 235) rats were excluded due to incorrect cannula placement. Anti-CIM6P/IGF2R antibody (R and D Systems # AF2447, Minneapolis, MN, USA) or IgG control was dissolved in PBS and injected at the indicated times at 5 or 50 ng/μL as described. At a concentration of 5 ng/μL, the antibody blocked 95% of CIM6P/IGF2R in an in vitro binding assay (R and D Systems). Puromycin (Sigma, St. Louis, MO, USA) was dissolved in PBS and co-injected (10 μg/side) with anti- CIM6P/IGF2R antibody or IgG, 15 min before IA training. Mannose-6-phosphate (Sigma) was dissolved in PBS and injected immediately after IA training at 5 μM, 5 mM, 25 mM, or 150 mM to generate a dose-response curve.

## Viral injections in mice

Mice were anesthetized with ketamine (75 mg/kg, i.p.) and xylazine (10 mg/kg, i.p.). The skull was exposed, and holes were drilled in the skull bilaterally above dHC. A Hamilton syringe with a 32-gauge needle mounted on a nanopump (KD Scientific, Holliston, MA) was stereotactically inserted into dHC (1.7 mm posterior to bregma, 1.5 mm lateral from midline, and 1.55 mm ventral from dura). AAV-DJ-hSyn-Cre-GFP or AAV-DJ-GFP ($5.8 \times 10^{12}$ genomic copies/mL, 1 μL per side; Gene Vector and Virus Core, Stanford University) was microinjected at a rate of 0.2 μL/min. The needle was left in place for an additional 5 min following microinjection to ensure complete diffusion of the AAV, and then slowly retracted. The scalp was sutured, and meloxicam (3 mg/kg, s.c.) was

administered as an analgesic treatment. Mice were returned to their home cage for 2 weeks to recover from the surgery and to allow viral expression.

## Subcutaneous injections in mice

Recombinant mouse IGF2 (R and D Systems, # 792 MG) was dissolved in 0.1% bovine serum albumin in PBS (BSA-PBS), and 30 µg/kg was injected subcutaneously. Mannose-6-phosphate (M6P) (Sigma, #M3655) was dissolved in PBS and 850 µg/kg was injected subcutaneously. Vehicle injections consisted of 0.1% BSA-PBS.

## Rat inhibitory avoidance (IA)

IA experiments were carried out as described previously (*Chen et al., 2011*; *Ye et al., 2017*). The IA chamber (Med Associates, St. Albans, VT, USA) consisted of a rectangular Plexiglas box divided into a safe compartment and a shock compartment. The safe compartment was white and illuminated by a light fixture on the compartment wall. The shock compartment was black and unilluminated. Footshock was delivered to the grid floor of this chamber via a constant-current scrambler circuit. The two compartments were separated by an automatically operated sliding door. During training sessions, each rat was placed in the safe compartment with its head facing away from the door. After 10 s (s) the door automatically opened, allowing the rat access to the shock chamber. The door closed 1 s after the rat entered the shock chamber, and a brief, 2 s footshock (0.6 mA or 0.9 mA, as indicated) was administered. Latency to enter the shock compartment was taken as a measure of acquisition. The rat was then returned to its home cage. Retention tests were performed at the indicated times by placing the rat back into the safe compartment and measuring the latency to enter the shock compartment. Footshock was not administered on the retention test, and testing was terminated at 900 s and performed by an experimenter blinded to the treatments given. The reminder shock (RS) procedure was used to test whether impaired memory could be reinstated. RS consisted of giving the animals a 2 s footshock of the same intensity as that received during training in a different context, which consisted of a square chamber (Med Associates) with three transparent walls, one opaque Plexiglas wall, and a floor grid with narrower spacing in a separate, well-lit room. Untrained control rats were handled, but otherwise remained in their home cage.

## Rat open field

Rats were allowed to freely explore an open field arena (75 × 75 × 20 cm$^3$) for 10 min, and their movements automatically tracked using EthoVision-XT (Noldus Information Technology, Wageningen, Netherlands). Locomotion was assessed using measures of total distance travelled (cm), as well as average velocity (cm/s). The arena was divided into 16 quadrants, and the time spent (s) in the four center quadrants were taken as time spent in the center, which is generally used as a measure of anxiety.

## Mouse open field

Mice were allowed to freely explore an open field arena (21 × 21 × 15 cm) for 5 min, and their movements automatically tracked using EthoVision-XT software. Locomotion was assessed using measures of total distance travelled (cm), as well as average velocity (cm/s). The arena was divided into 16 quadrants, and the time spent (s) in the four center quadrants were taken as time spent in the center, which is generally used as a measure of anxiety.

## Mouse novel object location (nOL)

Mice were trained on novel object location (nOL) by placing them into the arena that contained two identical objects (Mega Bloks 120, Quebec, Canada), and were permitted to freely explore for 5 min. Four hours later, mice were placed back into the area after one object had been moved to an alternate location. The other object remained in a constant location for both training and testing. Memory was measured as the percentage of time spent interacting with the object in the new location over the 5 min session. Animal behavior was video-recorded and analyzed off-line by an experimenter blind to the viral treatment.

## Mouse Pavlovian fear conditioning

The conditioning chamber consisted of a rectangular Perspex box (30.5 × 24.1 × 21.0 cm) with a metal grid floor (Med Associates) through which footshocks were delivered via a constant-current scrambler circuit. Freezing, defined as lack of movement besides heartbeat and respiration, was recorded every tenth second by a trained experimenter blind to the experimental conditions. The percentage of time spent frozen across the total number of observations was calculated (*Schrick et al., 2007*). The Pavlovian fear conditioning procedure was modified from previous studies (*Lovett-Barron et al., 2014*; *Stern et al., 2014*; *Zhang et al., 2005*). Mice were placed in the conditioning chamber; after 2 min they received one tone–shock pairing (30 s tone co-terminating with a 2 s, 1 mA footshock), and 30 s later they were returned to the home cage. Memory for the training context was tested by placing the mouse back into the conditioning chamber (1 or 7 days after training) for 3 min, in the absence of a footshock. Memory for the tone was tested in an alternate context (2 or 8 days after training). During tone testing, after 1 min in the new context (pre-tone), the tone was played for 2 min. Videos acquired during the 3 min context test, 1 min pre-tone, and 2 min tone were scored by an experimenter blind to the experimental procedures.

Mouse contextual fear conditioning Contextual fear conditioning was conducted similarly to Pavlovian fear conditioning. The procedure only differed in that the 2 s footshock during training was not signaled by a tone. Memory testing was carried out in the training context as described above, at the indicated timepoints after training. In experiments studying memory enhancement, contextual fear conditioning training was carried out using a lower footshock intensity, 0.7 mA, to prevent a ceiling effect in the freezing response.

## Immunofluorescence

Rats or mice were anesthetized with chloral hydrate (750 mg/kg) and transcardially perfused with ice-cold PBS (pH 7.4) followed by 4% paraformaldehyde (PFA) in PBS. Brains were isolated and post-fixed overnight in 4% PFA at 4°C, and then submerged in 30% sucrose in PBS for 72 hr. Brain sections (20 um) were cut using a cryostat for free-floating immunofluorescence staining. Sections were subjected to heat-mediated antigen retrieval in nanopure $H_2O$ for 2 min at 100°C. The sections were blocked with 10% normal goat serum, 3% bovine serum albumin, and 0.25% Triton X-100 in PBS for 2 hr at room temperature, and then incubated with primary antibodies in 0.25% Triton X-100 in PBS overnight at 4°C. The following primary antibodies were used: chicken anti-GFAP (1:5000, Abcam, #ab4674, Cambridge, MA, USA), rabbit anti-Iba1 (1:5000, Wako, #019–19741, Osaka, Japan), mouse anti-MAP2 (1:5000, Millipore, #MAB3418, Billerica, MA), mouse anti-CaMKIIα (1:1000, Millipore, #05–532), rabbit anti-IGF2R (1:1000, Abcam, #ab124767), chicken anti-GFP (1:1000, Aves Labs, #GFP-1020, Tigard, OR), rabbit anti-Arc (1:2000, Synaptic systems #156 003, Göttingen, Germany), rabbit anti-Egr1 (1:1000, Cell Signaling Technology, #4153, Danvers, MA, USA), and rabbit anti-c-Fos (1:500, Cell Signaling Technology, #2250). Sections were then washed with PBS three times for 10 min, and then incubated in secondary antibody for 2 hr at room temperature. Secondary antibodies used were goat anti-mouse, anti-rabbit, or anti-chicken conjugated to Alexa Fluor 488 or 568 (1:1000, Invitrogen, Waltham, MA, USA). Brain sections were washed with PBS three times for 10 min, and then mounted with Prolong Diamond antifade mounting medium with 4′,6-diamidino-2-phenylindole (DAPI, Invitrogen). Two brain sections from each animal (one section per bregma, at approximately bregma −3.6 mm and −4.0 mm for rats, and −1.7 mm and −2.2 mm for mice from dorsal hippocampus) were used for each set of staining. Image acquisition and quantification analysis were conducted by an experimenter blinded to the experimental conditions. Four images per sub-region (CA1 and DG) were imaged per hemisphere for each animal, yielding a total of 16 images for each sub-region, per animal. Images were captured as 1 μm z-stacks using 10 steps on a TCS SP8 confocal microscope (Leica, Wetzlar, Germany) at 63X magnification. Images were processed and quantified using the ImageJ software (US National Institutes of Health). All images for each independent experiment were processed in the same manner: first, background was subtracted and a maximum projection image was rendered using ImageJ. To quantify the intensity and number of immunopositive cells, images were subjected to thresholding, fill holes, watershed, and analyze particles (*Hartig, 2013*; *Jedlicka et al., 2015*; *Roy et al., 2016*; *Vasileiou et al., 2016*). For each hippocampal subregion, sixteen images were analysed, and their values normalized against their total number of cells determined using DAPI (4′,6-diamidino-2-phenylindole) staining, a marker commonly

used to visualize nuclei. The values per sub-region per each animal were averaged. The data per group of animals in each experimental condition were expressed as intensity or number of positive cells (in %), both normalized against the total number of cells (DAPI staining). For co-localization analysis, single z planes were extracted from the z-stack. To visualize dorsal hippocampus, multiple overlapping images (10% overlap) were captured at 20 × magnification, and a composite image was constructed using the LAS X software (Leica).

## RNA extraction and RT-PCR analyses

Rats were euthanized by decapitation. Their brains were quickly extracted; dorsal hippocampi from bregma −1.6 mm to −5.4 mm were rapidly dissected on ice using a brain matrix, and then submerged in Qiazol (Qiagen, Hilden, Germany). Total RNA was isolated using the RNeasy Plus Universal Mini Kit (Qiagen, #73404), and 250 ng of RNA was reverse transcribed using QuantiTect Reverse Transcription Kit (Qiagen, #205311). PCR amplification consisted of: initial denaturation at 95°C for 5 min, followed by 40 cycles of 94°C for 30 s, 60°C for 30 s, 72°C for 20 s, and a final extension step at 72°C for 10 min. Quantitative real-time PCR analysis was done using CFX96 Touch Real-Time PCR Detection System (Bio-Rad, Hercules, CA, USA) with iQ SYBR Green Supermix (Bio-Rad, #107–8882). See Key Resources Table for primer sequences.

Three technical replicates were run for each sample, and the average cycle threshold (Ct) value was used for quantification using the relative quantification method. Ct values for genes of interest were normalized against the corresponding values for *Gapdh*. Values for each animal were expressed as percentage of the value for the control group (as specified in each experiment untrained, or IgG-injected-untrained).

## Whole and synaptoneurosomal protein extracts and western blot analysis

Rats were euthanized by decapitation. Their brains were quickly extracted and dorsal hippocampi from bregma −1.6 mm to −5.4 mm were rapidly dissected on ice using a brain matrix. The tissue was snap-frozen on dry ice for total extracts. Tissue was homogenized in radioimmunoprecipitation assay (RIPA) buffer (150 mM NaCl, 1% Triton X-100, 0.5% sodium deoxycholate, 0.1% SDS, 5 mM EDTA, 10% glycerol, 50 mM Tris, pH 8.0) supplemented with 0.5 mM PMSF, 2 mM DTT, 1 mM EGTA, 1 µM microcystin LR, 10 mM NaF, 1 mM $Na_3VO_4$, benzamidine, protease inhibitor cocktail, and phosphatase inhibitor cocktails (Sigma). Homogenates were centrifuged at 21,300 *g* at 4°C for 30 min, and the supernatant was retained. Synaptoneurosomal extracts (SN) were prepared as previously described (*Chen et al., 2011*; *Steinmetz et al., 2018*). Freshly dissected hippocampi were homogenized with a glass–Teflon homogenizer in ice-cold buffer (10 mM HEPES, 2 mM EDTA, 2 mM EGTA, 0.5 mM DTT, phosphatase and protease inhibitor cocktails [Sigma]). Homogenates were then sequentially filtered through a 100 µm nylon mesh filter and a 5 µm nitrocellulose filter. To obtain SN fractions, the samples were centrifuged at 1000 *g* at 4°C for 10 min, and the pellets were resuspended in RIPA buffer.

Protein concentrations were determined using the Bio-Rad protein assay (Bio-Rad). Twenty micrograms of total protein extract was loaded per lane, resolved on denaturing SDS-PAGE gels, and transferred to Immobilon-FL membranes (Millipore). The membrane was dried and then with 5% BSA in Tris-buffered saline with 0.1% Tween 20 (TBST, pH 7.4). Membranes were incubated overnight in 4°C in primary antibody diluted in TBST.

The following primary antibodies were used: rabbit anti-IGF2R (1:1000, Abcam, #ab124767), rabbit anti-Egr1 (1:1000, Cell Signaling Technology, #4153), rabbit anti-Cre (1:1000, Cell Signaling Technology, #15036), and mouse anti-actin (1:10000, Santa Cruz Biotechnology, #sc-47778, Dallas, TX, USA). The membranes were washed three times in TBST for 10 min and incubated in secondary antibodies for 1 hr at room temperature. The following secondary antibodies were used: anti-rabbit IRDye800CW and anti-mouse IRDye680 (1:10000, Li-Cor, Lincoln, NE, USA). After three additional 10 min TBST washes, membranes were scanned on the Li-Cor Odyssey imager under non-saturating conditions. Data were quantified using pixel intensities with the Odyssey software (Li-Cor). Intensities were normalized against the corresponding intensities of actin immunoreactivity and expressed as percentages relative to the control group.

### *In vivo* SUnSET

Fifteen minutes prior to IA training, puromycin (10 µg) was co-injected with IgG or anti-CIM6P/ IGF2R bilaterally into the hippocampus of rats as described above. We used a protocol previously established in the laboratory that showed reliable detection of puromycin incorporation 2 hr after IA training (*Descalzi et al., 2019*). Rats were transcardially perfused as describe above, and brains were cryosectioned. Similar to the staining procedure described above, coronal sections underwent incubation with a blocking solution for 2 hr at room temperature, and then incubated with mouse anti-puromycin conjugated to Alexa Fluor 647 (1:1000, Millipore #MABE343) for 2 hr at room temperature. After washing three times with PBS for 10 min, the sections were mounted with Prolong Diamond antifade mounting medium with DAPI (Invitrogen), and imaged as described above.

### Statistical analyses

Data were analyzed using Prism 6 (GraphPad Software, San Diego, CA). The number of independent experiments carried out and the numbers of biological replicates [i.e., animals (n)] are indicated in each figure legend. Data are expressed as means ± standard error of the mean (S.E.M.). No statistical method was used to predetermine sample size. P values were generated using Student's t-tests, one- or two-way analysis of variance (ANOVA), or repeated measure (RM) ANOVA followed by Sidak or Tukey post hoc tests. All analyses were two-tailed, and data were considered significant when p<0.05. The numbers of subjects used in our experiments were the minimum required to obtain statistical significance, based on our experience with the behavioral paradigm and in agreement with standard literature. Both male and female transgenic mice were used for our experiments. Preliminary statistical analyses comparing males and females (n = 2–4) showed no significant difference in value distribution (unpaired two-tailed Student's t-test, p>0.05), therefore, males and females were combined into a single group for between group comparisons. Although these values of n are too low for any robust statistical analysis, we decided to group the subjects and to refrain from pursuing sex-related questions.

## Acknowledgements

We thank Dr. David Skaar (NC State University, North Carolina) for generously providing breeder *Igf2r*-floxed mice. The studies described in this manuscript were supported by NIH grant MH065635 to CMA.

## Additional information

### Funding

| Funder | Grant reference number | Author |
|---|---|---|
| National Institute of Mental Health | MH065635 | Cristina Alberini |

The funders had no role in study design, data collection and interpretation, or the decision to submit the work for publication.

### Author contributions

Xiao-Wen Yu, Data curation, Formal analysis, Investigation, Methodology, Writing - original draft; Kiran Pandey, Aaron C Katzman, Data curation, Formal analysis, Investigation, Methodology; Cristina M Alberini, Conceptualization, Data curation, Supervision, Funding acquisition, Investigation, Methodology, Writing - original draft, Writing - review and editing

### Author ORCIDs

Xiao-Wen Yu https://orcid.org/0000-0002-8974-0085
Kiran Pandey https://orcid.org/0000-0001-8658-2942
Cristina M Alberini https://orcid.org/0000-0001-7386-0018

## Ethics

Animal experimentation: All protocols complied with the National Institutes of Health Guide for the Care and Use of Laboratory Animals and were approved by the Institutional Animal Care and Use Committee at New York University (Public Health Service (PHS) Policy on Humane Care and Use of Laboratory Animals Assurance A3317-01 and AAALAC Assurance 00-1350 to NYU Washington Square campus). All surgery was performed under ketamine/xylazine anesthesia and meloxicam as described in the methods. Every effort was made to minimize suffering.

## Decision letter and Author response

Decision letter https://doi.org/10.7554/eLife.54781.sa1
Author response https://doi.org/10.7554/eLife.54781.sa2

## Additional files

### Supplementary files

• Source data 1. Detailed information and statistical analyses related to data presented in the manuscript.

• Transparent reporting form

### Data availability

All data generated or analysed during this study are included in the manuscript and supporting files. Information about sample-size estimation is provided in the Statiscal analysis sub-section of the Materials and Methods section.

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
