## [Decision Letter]

**Acceptance summary:**

This study uses a combination of biochemical and behavioral approaches to demonstrate a critical role for the CIM6P/IGF2 receptor in the hippocampus in long-term memory. The authors report that neuronal CIM6P/IGF2R is required for learning-induced de novo protein translation and works as a target mechanism for memory enhancement. The authors provided a thoughtful and detailed response in regards to previous reviewer comments. The previous concerns included a request for an additional experiment, which the authors addressed based on the current literature and the confounds associated with the inability to do additional experiments during COVID-19. Given eLife policy regarding the current shutdown, the authors' detailed response is sufficient.

**Decision letter after peer review:**

Thank you for submitting your article "A role for CIM6P/IGF2 receptor in memory consolidation and enhancement" for consideration by eLife. Your article has been reviewed by three peer reviewers, including Lisa M Monteggia as the Reviewing Editor and Reviewer #1, and the evaluation has been overseen by Gary Westbrook as the Senior Editor. The following individual involved in review of your submission has agreed to reveal their identity: Kobi Rosenblum (Reviewer #2).

The reviewers have discussed the reviews with one another and the Reviewing Editor has drafted this decision to help you prepare a revised submission.

Summary:

This study uses IHC, behavior, biochemistry, genetics and molecular research to show the role of IGF2 receptor specifically in memory consolidation. They show that CIM6P/IGF2R receptor has an important role in hippocampus-dependent memory consolidation in rats and mice. Blocking CIM6P/IGF2R receptor 15 min before fear conditioning training impaired memory consolidation. In addition, using pharmacological and genetic approaches, the authors show that CIM6P/IGF2R receptor controls de novo protein synthesis and up regulation of Arc, Egr1 and c-Fos proteins following fear condition training. Moreover, administration of mannose-6-phosphate/IGF2, CIM6P/IGF2R ligands, enhance memory retention which was occluded by CIM6P/IGF2R receptor cre induced expression in the dorsal hippocampus in mice. This is a well carried out study that shows IGF2R is required for hippocampal memory consolidation in mice and rats.

This paper is an extension of the authors previous work on IGF2 and suggests that IGF2R in the hippocampus may be more important than IGF2 in memory. Although this is a strong study that convincingly shows IGF2R is required for hippocampal memory consolidation there were concerns that should be addressed.

Major comment:

All three reviewers raised the question regarded how the authors link IGF2R and the downstream IEG translation. Is there any mechanistic insight into how IGF2R links to IEG translation?

Comments:

1) The authors show the manipulation does not affect retrieval. Does it affect reconsolidation?

2) Measuring intensity in the IHC can be problematic. Do the authors have data for n of cells? Can they use an internal control (not relevant brain structure)?

3) Can the authors speculate why the window of IGF2R engagement is so long? It seems unlikely that circulating ligands are present for 8hrs after learning. Indeed, blocking IGF2R leads to rapid memory impairments (5 min-1hr after learning). To clarify this point, the authors could inject the antibody immediately and 5 min or 1hr after training.

4) This rapid engagement of IGFR2 is in contrast to the authors previous work (Chen et al., 2011) showing that IGF2 expression is only increased 20hrs after avoidance training in rats. No increase was observed prior to that time point. Is this because other ligands are important immediately after learning in the normal context?

5) Subsection “Neuronal CIM6P/IGF2R is required for the formation of hippocampus-dependent memories”. Figure 4A is not knock down data, this is Figure 4G-I.

6) Discussion, eighth paragraph, last sentence, please correct.

7) In figure legend please specify rat or mice according to the experiment.

8) It is not clear which subregions are targeted in the stereotaxic injections of rats and mice as the coordinates are missing in the "Materials and methods".

9) Figure 1 – western blot data in Figure 1C, D. Figure 1C shows one upper band and a lower band ~ 250kDa, while Figure 1D shows two upper bands without the lower band ~ 250kDa. It is not clear which band is the real signal, raising concern about the specificity of the antibody in rat tissue although it seems to be specific in mouse tissue.

10) Figure 2: Is the inference that the single injection is ineffective because the receptor is already engaged sufficiently? Would it be useful to have this be measured somehow?

11) Figure 3: Are we talking about the same cohort of animals across all panels?

12) Figure 4F the authors see impaired contextual memory at 5 min but not 1 min after the training. It is not clear how to define these two processes temporally. Could the freezing level of mice 1 min after training be due to the response to the shock but not the index of learning? The authors state n=8 for the IHC data. Given that they further state in the Materials and methods 2 slices/animals, over a range of 0.4mm in rats and 0.5mm in mice, this should be defined more specifically.

13) Figure 5: The IHC data here needs to account for the range over which it was collected over, in order to be comparable to mRNA studies, conducted with extracted/dissected hippocampi.

Are there any differences in the number of positive cells for the protein products of these IEGs?

14) Figure 6: it was shown previously in Figure 3C in rats that there is significant effect of anti CIM6P/IGF2R on memory tested 1h after training, so why in in Figure 6 protein synthesis using puromycin was measured 2 h following inhibitory place avoidance training (subsection “CIM6P/IGF2R is required for the training-induced increase in de novo synthesized protein”).

15) Figure 7: Considering the dose response obtained with M6P in rats, and the lack of an effect at the high dose, this might be particularly revealing of the potential and limitations associated with the ligand. Does the high dose cause defects in physiology or any other memory-related paradigms?

---

## [Author Response]

[…]Major comment:All three reviewers raised the question regarded how the authors link IGF2R and the downstream IEG translation. Is there any mechanistic insight into how IGF2R links to IEG translation?

We agree with the reviewers that this is one of the most important next questions that should be addressed. We are working extensively towards the identification of the mechanisms directly controlled by CIM6P/IGF2R.

The current knowledge on the mechanisms of action of CIM6P/IGF2R has derived mostly from studies carried out on tumor cell lines and embryonic development, which then have been corroborated by knockout mice investigations. These studies identified a role for CIMPR/IGF2R in trafficking certain ligands such as IGF2 and lysosomal enzymes (mannose-6-phosphate tagged proteins) to and from endosomal compartments, lysosomes, Golgi apparatus, and/or plasma membrane.

To our knowledge, ours is the first study that has examined the role of CIM6P/IGF2R in the healthy brain and has identified its crucial function in memory formation. Because CIM6P/IGF2R has a small intracellular domain and does not have known direct signal transduction mechanisms (El-Shewy and Luttrell, 2009), hypothesizing molecular mechanisms directly activated downstream is not straightforward.

In our study, we therefore approached the question of CIM6P/IGF2R downstream mechanisms by testing whether the receptor controls fundamental and rapidly engaged cellular processes (close to plasma membrane) that are known to be promptly recruited by learning and required for long-term memory. Hence, we assessed whether the receptor regulates activity-induced de novo translation. Using in vivo SUnSET, we found that de novo translation evoked by learning is, in fact, downstream of the receptor; blocking the receptor completely blocked learning-induced translation.

The effect on translation was then extended to the translation of specific proteins, i.e., IEGs. Blockade of the receptor did not affect the induction of IEG transcription upon learning, but completely blocked their translation. These results provided key information on mechanisms downstream of the CIM6P/IGF2R and suggested that our future studies should focus on mechanisms that link endosomal trafficking, lysosomal degradation and Golgi functions to learning-induced de novo translation.

This is a novel and complex area of investigation, and while we are devoting many resources to answering this important question, finding the answer will require multiple approaches (in vitro and in vivo, targeted and unbiased) and will not be a matter of months. Given the significant amount of work involved, we feel that this aim is outside the scope of the current manuscript and hope that the reviewers agree.

Comments:1) The authors show the manipulation does not affect retrieval. Does it affect reconsolidation?

In Figure 3E, we show that injecting the blocking antibody 15 min before memory test (retrieval) given at 24 hours after training had no effect on that test, hence on memory retrieval. We then tested again the memory of those animals 7 days later, and again found no effect on memory retention. These data indicated that reconsolidation was not affected. Like what we found for consolidation, where a pre-training injection completely blocked memory, if CIM6P/IGF2R was required for the rapidly induced fragility and restabilization of memory upon retrieval, i.e. reconsolidation, pre-retrieval injection should have impaired memory at the 7 day test. In our previous submission we did not emphasize these results because the paper focuses on the consolidation process.

Now we included the explanation of the no-effects at the 7 days test after the pre-retrieval injection of the receptor blocking antibody, hence the lack of an effect on reconsolidation, in the Abstract, Results, and Discussion sections.

2) Measuring intensity in the IHC can be problematic. Do the authors have data for n of cells? Can they use an internal control (not relevant brain structure)?

To address this question, we have now re-analyzed our intensity data in order to normalize them against the total number of cells present in the quantified samples.

For each image, we quantified the total number of cells using DAPI (4′,6-diamidino-2-phenylindole) staining, a marker commonly used to visualize nuclei. As we have now detailed in the Materials and methods section, each image was subjected to background subtraction, thresholding, fill holes, watershed, and the cells were quantified using analyze particles (Hartig, 2013; Roy et al., 2016; Jedlicka et al., 2015; Vasileiou et al., 2016). The number of DAPI-positive cells for all images per sub-region were summed to calculate the total number of cells/sub-region, per rat. For each rat, 1100 – 1600 cells were quantified in CA1, and 1900 – 3000 cells in DG. The total number of cells per rat in each experimental group (effects of training and treatment) underwent statistical analyses to determine if changes among the groups were found. No difference in the averaged total number of DAPI-positive cells was found across training and/or treatment groups, in either sub-region (these data are depicted in a new figure: Figure 5—figure supplement 1).

Then, we re-analyzed the immunofluorescence total intensity for each IEG, in both CA1 and DG, by normalizing the raw integrated density values against the total number of DAPI-positive cells for that sub-region (graphs in Figure 5B-D). This new analysis showed that training significantly increases the intensity of all three IEGs, and that the receptor-blocking antibody blocked this increase. The graphs reporting these data have been included in the updated Figure 5 and the new statistical analyses is shown in the Supplementary file 1.

In addition, on the same images, we quantified the number of IEG-positive cells, normalized this number against the total number DAPI-positive cells for each rat, and expressed the results in % change for each sub-region/rat. Statistical analyses across the grouped values determined changes related to training and/or treatment. As shown in the updated Figure 5 (statistical analyses in Supplementary file 1), training significantly increased the percent of Arc-positive cells in both CA1 and DG. The same effect was observed in DG for Egr1 and c-Fos in DG but not in CA1 where the % of positive cells did not change (but the intensity did as described above). All these changes were blocked by the receptor-blocking antibody.

To re-summarize, training significantly increased both the number of Arc-positive cells, as well as Arc total intensity. However, in the case of Egr1 and c-Fos, training differentially affected the two hippocampal sub-regions: while it significantly increased Egr1 and c-Fos cell number and intensity in DG, it only increased Egr1 and c-Fos intensity but not the number of positive cells in CA1, indicating that cells already expressing Egr1 and c-Fos upregulated their levels. All these training-induced changes required CIM6P/IGF2R, as they were completely blocked by the receptor-blocking antibody.

These findings are now described in the Results section; their statistical analyses are detailed in Supplementary file 1, and the related methods have been added to the Materials and methods section.

We think that using a putative “non relevant” brain region, presumably not involved in inhibitory avoidance memory consolidation, is most likely not the best control, because changes in IEGs are not selective for memory but occur in response to a variety of brain activations. These changes can occur for instance in response to sensory and motor stimuli. In other words, such an experiment would be informative only if no changes are found in the putative “non relevant” area.

3) Can the authors speculate why the window of IGF2R engagement is so long? It seems unlikely that circulating ligands are present for 8hrs after learning. Indeed, blocking IGF2R leads to rapid memory impairments (5 min-1hr after learning). To clarify this point, the authors could inject the antibody immediately and 5 min or 1hr after training.

We know from previous studies that, in the rat hippocampus, the phase of inhibitory avoidance consolidation that requires transcription starts very rapidly with training and then persists for more than 24 hours; finally returning to control conditions by 2 days after training (e.g. Taubenfeld et al., 2001; Bambah-Mukku et al., 2014; Garcia-Osta et al., 2006; Chen et al., 2011; Arguello et al., 2013).

Therefore, it is not surprising that the requirement for CIMPR/IGF2R during memory consolidation can extend for several hours. Circulating ligands could be engaged for the entire consolidation temporal window, but, even if they were involved for a more limited time, they could functionally influence subsequent processes for an extended time.

Toward the delineation of the temporal window of receptor requirement, we have carried out a relatively extensive temporal and concentration testing (shown in Figure 1). These experiments revealed that inhibition of the receptor acutely prior to training completely blocks memory formation; however, the receptor needs to be blocked for several hours after training, through multiple injections, in order to show its critical role during consolidation. In fact, even an acute injection of a 10 times higher concentration of the blocking antibody immediately after training has no effect on memory. Hence, these experiments indicated that training very rapidly engages the receptor, which then remains functionally recruited for an extended time.

In normal circumstances, we would have carried out one of the experiments suggested by the reviewer, and specifically testing the effect of 2 injections post-training separated by one hour. We believe that the other experiment of testing a double injection with the first one delivered immediately after training and the second 5 minutes later, is a condition already addressed by the single post-training injection at 10x higher concentration (no effect). However, given the uncertainty of when the experiment could be done due to the COVID-19 situation and the fact that our tests have already provided an extensive temporal investigation, we decided to clarify our results and add a discussion point (as detailed below).

We think that the additional experiment of 2 injections separated by one hour would only perhaps confirm that a shorter temporal window between 2 injections is also effective. We hope that the reviewer will be satisfied. However, if the reviewer feels that we should perform this experiment, we will do so after the COVID resolution, but we need to request an extension for our resubmission until when we will be allowed to return to the laboratory.

To summarize, a single injection given either before training or immediately after training, or at 8 hours after training, or even a single post-training injection at 10x concentration, along with double injections given 8 hours apart, showed that, once it is rapidly recruited by training, the CIM6P/IGF2R needs to remain functionally engaged for several hours. The mechanisms by which the receptor is functionally recruited for several hours following training remain to be identified. However, here we offer some speculations: 1) Our data based on in vivo SUnSET indicated that activation of CIMPR/IGF2R is upstream of learning-induced mRNA translation. Previous studies have shown that, in order for memories to be successfully consolidated, multiple waves of de novo mRNA translation are required (Barzilai et al., 1989; Bourtchouladze et al., 1998; Taubenfeld et al., 2001; Alberini, 2009; Stork and Welzl, 1999). Moreover, as mentioned, the requirement for de novo translation in the hippocampus following IA continues for more than 24 hours after training (Bambah-Mukku et al., 2014; Bekinschtein et al., 2007). Hence, if, as suggested by our data, CIM6P/IGF2R is temporally coupled to de novo translation, it is plausible that its functional engagement follows the same temporal window of requirement.

Indeed, a pre-training injection of the receptor blocking antibody, like that of translation inhibitors (e.g. anisomycin) completely block memory formation, whereas post-training injections of either blocking antibodies or protein synthesis inhibitors require more extended treatments (Bambah-Mukku et al., 2014; Milekic et al., 2006).

To address the reviewers’ question, we have now added these clarifications and speculations in the Discussion section.

4) This rapid engagement of IGFR2 is in contrast to the authors previous work (Chen et al., 2011) showing that IGF2 expression is only increased 20hrs after avoidance training in rats. No increase was observed prior to that time point. Is this because other ligands are important immediately after learning in the normal context?

In our previous work (Chen et al., 2011), we found that mRNA levels of IGF2 in the hippocampus did not change for several hours after IA training, but significantly increased at 20 hours post-training and then returned to control levels by 48 hours after training. IGF2 protein levels followed the same profile. Lack of protein induction in response to training, however, does not exclude that that protein, which is already present at a lower level at the time of training, is readily functionally engaged. Changes in expression may be additionally needed to either sustain the protein function during the temporal window of consolidation (effect on persistence) and/or to replenish the consumed protein.

A similar condition has been found with brain-derived neurotrophic factor (BDNF). BDNF is rapidly engaged through training and immediately required for the formation of long-term memory without rapidly changing its protein levels. Its level in fact significantly augments only at later timepoints, and this late increase, which is part of a positive autoregulatory loop, is required for memory consolidation (Bambah-Mukku et al., 2014).

Thus, IGF2 could be readily engaged via CIM6P/IGF2R at the time of training, and then increase its expression level at later time points, while continuing to be functionally required (Chen et al., 2011).

Nevertheless, we cannot exclude, and agree with the reviewers’ suggestion, that at different timepoints after training, the receptor may engage other ligands in physiological conditions. In fact, it is known that CIM6P/IGF2R can bind a variety of ligands, including mannosylated proteins such as lysosomal enzymes, TGF-β1 precursor, proliferin, plasminogen, and retinoic acid. We have added a discussion point on this issue (Discussion).

5) Subsection “Neuronal CIM6P/IGF2R is required for the formation of hippocampus-dependent memories”. Figure 4A is not knock down data, this is Figure 4G-I.

This has been corrected.

6) Discussion, eighth paragraph, last sentence, please correct.

This sentence has been re-worded to make this point clearer.

7) In figure legend please specify rat or mice according to the experiment.

We have now clarified whether the experiment was conducted in rats, mice, or both in each figure legend.

8) It is not clear which subregions are targeted in the stereotaxic injections of rats and mice as the coordinates are missing in the "Materials and methods".

All injections in rats targeted dorsal hippocampus. We detailed the rat coordinates for cannula placement and injection in the “Materials and methods” sub-section “Cannula implants and hippocampal injections in rats”. The coordinates of stereotactic injections of viruses in mice are detailed in the “Materials and methods” sub-section “Viral injections in mice”.

9) Figure 1 – western blot data in Figure 1C, D. Figure 1C shows one upper band and a lower band ~ 250kDa, while Figure 1D shows two upper bands without the lower band ~ 250kDa. It is not clear which band is the real signal, raising concern about the specificity of the antibody in rat tissue although it seems to be specific in mouse tissue.

We apologize for any confusion in the western blot data. Both Figure 1C and D are data from rat lysates; however, the tissue processing was not identical, as part of the goal in Figure 1C was to compare the synaptoneurosomal and total fractions from untrained and trained rats. In the experiment shown in Figure 1D, the tissue was dissected and snap frozen, then later homogenized in RIPA buffer for protein extraction. In Figure 1C, the tissue was freshly dissected and homogenized in synaptoneurosomal buffer (see Materials and methods), in order to allow separation of synaptosomes. An aliquot of this suspension was extracted in RIPA buffer to generate the total input protein extract. It is possible that the difference in lysate preparation procedures may lead to the slightly different appearance in the bands.

Nevertheless, it is established that the CIM6P/IGF2R molecular weight is about 300 kDa. The 300 kDa band is the strongest band (can appear as a doublet relative to the separation) evident with either type of extraction protocols, and well above 250 kDa. This reactivity has been verified by Abcam to be blocked by CIMPR/IGF2R peptides. In our experiments we have quantified the strong 300 kDa signal. The lower band in Figure 1C could be a non-specific reactivity.

Notably, a similar 300 kDa band was also detected in total protein extracts obtained from mouse dorsal hippocampus (homogenized in RIPA buffer; Figure 4G). We have validated the specificity of this 300 kDa band with genetic knockdown: as shown in Figure 4G, the 300 kDa band was significantly decreased in homogenates obtained from mice that underwent Cre-recombinase-dependent knockdown of CIM6P/IGF2R.

10) Figure 2: Is the inference that the single injection is ineffective because the receptor is already engaged sufficiently? Would it be useful to have this be measured somehow?

Please see also response above to point 3. Our results indeed indicate that the timing of receptor engagement is key. When comparing a single injection given 15 min before training (Figure 2F) to one given immediately after training (Figure 2B), the outcome is radically different: the former completely blocks memory formation, while the latter has no effect. This suggests that the receptor is recruited very rapidly with training and very quickly activates downstream mechanism(s). The few minutes after training, that elapse before the antibody blocks the receptor, are sufficient for the receptor itself to be engaged sufficiently. Once engaged by the training event, the receptor seems to remain engaged for several hours, as a single injection no longer affects memory retention, but a double injection, hours apart, does. We do not know of methods available other than blocking the receptor (with the blocking antibody) to efficiently measure receptor engagement.

11) Figure 3: Are we talking about the same cohort of animals across all panels?

Independent cohorts of rats were used for each lettered panel in Figure 3, in order to avoid the confound of multiple testing. We have clarified this point in the text.

12) Figure 4F – the authors see impaired contextual memory at 5 min but not 1 min after the training. It is not clear how to define these two processes temporally. Could the freezing level of mice 1 min after training be due to the response to the shock but not the index of learning? The authors state n=8 for the IHC data. Given that they further state in the Materials and methods 2 slices/animals, over a range of 0.4mm in rats and 0.5mm in mice, this should be defined more specifically.

We do not interpret the freezing at 1 min as the result of shock only; rather, we think that our experiments show a very rapid effect on blocking the CIM6P/IGF2R on memory retention. In several studies, in fact, memory retention is tested immediately after training (e.g. Radulovic et al., 1998; Balogh et al., 2003). We are not aware of any work indicating that associative memory requires time to be expressed.

As the receptor is known to be involved in endosomal trafficking, the rapid mechanism could be, for example, vesicle trafficking and/or local mRNA translation. In line with the latter idea, our results showed that pre-training receptor blockade completely disrupts the de novo translation induced by learning, a process known to occur very rapidly. We added a discussion point (Discussion).

We apologize for any confusion regarding the coordinates for IHC sections. We have edited the text in the Materials and methods section, to clarify that the sections were taken at the two specified coordinates from each animal.

13) Figure 5: The IHC data here needs to account for the range over which it was collected over, in order to be comparable to mRNA studies, conducted with extracted/dissected hippocampi.Are there any differences in the number of positive cells for the protein products of these IEGs?

The coordinates for both IHC and RNA extracted are the dorsal hippocampus, the focus of our studies. The RNA was extracted from the entire dHC. For the IHC analyses we quantified 1 section at approximately Bregma -3.6 mm and another at -4.0 mm for each animal on each side of the dorsal hippocampus. These sections provide an average of sample-changes along the dHC.

As detailed above in point 2, we quantified the number of IEG-positive cells, normalized this number against the total number DAPI-positive cells in each rat, and expressed the results as a percent of positive cells for each sub-region. As shown in the updated Figure 5, training significantly increased the percentage of Arc-positive cells in both CA1 and DG. This effect was blocked in the rats that received the receptor-blocking antibody. The same effect was observed in DG for Egr1 and c-Fos. However, training did not change the percentage of Egr1- or c-Fos-positive cells in CA1(but did change the intensity of expression). Graphs depicting this new analysis have been added to Figure 5, and their statistics added to Supplementary file 1.

All these training-induced changes required CIM6P/IGF2R, as they were completely blocked by the receptor-blocking antibody.

These findings are now described in the Results section; their statistical analyses are detailed in Supplementary file 1, and the related methods have been added to the Materials and methods section.

14) Figure 6: it was shown previously in Figure 3C in rats that there is significant effect of anti CIM6P/IGF2R on memory tested 1h after training, so why in in Figure 6 protein synthesis using puromycin was measured 2 h following inhibitory place avoidance training (subsection “CIM6P/IGF2R is required for the training-induced increase in de novo synthesized protein”).

With the in vivo SUnSET assay measured with IHC analysis we did not intend to correlate the result of puromycin incorporation with the 1 hour memory impairment, but aimed at determining the effect of receptor blockade on the level of de novo translation. We had established conditions in the lab working reliably for the in vivo SUnSET assessment at 2 hours after training; hence, we used that timepoint.

15) Figure 7: Considering the dose response obtained with M6P in rats, and the lack of an effect at the high dose, this might be particularly revealing of the potential and limitations associated with the ligand. Does the high dose cause defects in physiology or any other memory-related paradigms?

It is very common in pharmacology that dose-response curves of compounds show optimal concentration effects and then higher concentrations produce impairments. Usually these types of curves indicate that higher doses may have other effects. In fact, we found similar outcomes with systemic IGF2 injections: mice given a systemic injection of IGF2 only exhibited enhanced memory for contextual fear conditioning within a given concentration range, while lower and higher doses did not elicit memory enhancement (Stern et al., 2016a).

We did not detect any overt deficits in rats injected with the high dose of M6P, however, as stated in the manuscript, the dose was injected only once.

References:

Balogh, S.A., Radcliffe, R.A., Logue, S.F., Wehner, J.M., 2003. Contextual and cued fear conditioning in C57BL/6J and DBA/2J mice: Context discrimination and the effects of retention interval. Behav. Neurosci. 116, 947–57. doi:10.1037/0735-7044.116.6.947

Bekinschtein, P., Cammarota, M., Igaz, L.M., Bevilaqua, L.R.M., Izquierdo, I., Medina, J.H., 2007. Persistence of Long-Term Memory Storage Requires a Late Protein Synthesis- and BDNF- Dependent Phase in the Hippocampus. Neuron 53, 261–277. doi:10.1016/j.neuron.2006.11.025

Milekic MH, Brown SD, Castellini C, Alberini CM. (2006) Persistent disruption of an established morphine conditioned place preference. J Neurosci. 2006 Mar 15;26(11):3010-20.

Radulovic, J., Kammermeier, J., Spiess, J., 1998. Generalization of fear

responses in C57BL/6N mice subjected to one-trial foreground contextual fear conditioning. Behav Brain Res 95, 179–189. doi:10.1016/S0166-4328(98)00039-4.